

# Optical Properties of North Atlantic Aerosols Through a compact dual-wavelength depolarization Lidar Observations

Yenny González[1,2], María F. Sánchez-Barrero[3], Ioana Popovici[1,3], África Barreto[2,4], Stephane Victori[1], Ellsworth J. Welton[5], Rosa D. García[6,2], Pablo G. Sicilia[2], Fernando A. Almansa[1,2], Carlos Torres[2], Philippe Goloub[3]

[1]Scientific department, CIMEL Electronique, Paris, 75011, France
[2]Izaña Atmospheric Research Centre (IARC), State Meteorological Agency (AEMET), Santa Cruz de Tenerife, 38001, Spain
[3]Univ. Lille, CNRS, UMR 8518 - LOA - Laboratoire d'Optique Atmosphérique, F-59000 Lille, France
[4]Atmospheric Optics Group of Valladolid University (GOA–UVa), Valladolid University, Valladolid, Spain
[5]NASA Goddard Earth Sciences and Technology Center, University of Maryland, Baltimore County, Baltimore, MD 21250.
[6]TRAGSATEC, Madrid, Spain.

*Correspondence:* Yenny González (y-gonzalez@cimel.fr)

**Abstract.** We present a characterization of the optical properties of different aerosol types based on data collected using a compact dual-wavelength depolarization elastic lidar (532 and 808 nm, CIMEL CE376). We evaluate the vertical distribution and temporal evolution of a variety of aerosols observed in the subtropical North Atlantic region, covering from Saharan dust outbreaks and volcanic aerosols to fresh and aged wildfires, observed between August 2021 and August 2023, using a modified two-wavelength Klett inversion method to derive the aerosol backscatter and extinction coefficients from CE376 lidar observations. We assessed the performance of the CE376 lidar during an intercomparison campaign with an MPL-4B lidar (MPLNET) while collocated at the facilities of the Izaña Atmospheric Research Centre (Canary Islands, Spain). Both instruments depicted the vertical aerosol structure similarly. The main differences were attributed to errors arising from the determination of the overlap function and the depolarization calibration in each instrument and the larger effect of the solar background on the CE376 system during daylight. Absolute differences in the volume depolarization ratio ($\delta^v$) were 0.3% reduced to 0.2% when only nighttime data was used. The measurements of particle linear depolarization ($\delta^p$), extinction Ångström exponent (EAE (532/808)) and attenuated colour ratio (ACR (808-532)), provided by the combination of the two channels of the CE376, allow us to describe the composition and size of the studied aerosols. Fresh Saharan dust particles were the largest particles found in this study with non-spherical morphology and traveling in a well-mixed layer, exhibiting the lowest EAE (532/808) and highest ACR (808-532) and $\delta^p$ (532) > 0.15. The smaller particles with quasi homogeneous morphology were attributed to sulphate aerosol from the initial stage of the Cumbre Vieja volcano eruption and aged Canadian wildfires traveling across the Atlantic, showing the lowest $\delta^p$ (< 0.08) and the highest EAE (532/808) (>1). Middle values of these retrieved parameters were associated with the heterogenous mixture of ash, soot, and charred vegetation from fresh local forest wildfires. The retrieved properties demonstrate the excellent performance of the CE376 micro-lidar as a tool for continuous monitoring and characterization of the temporal and vertical distribution of atmospheric aerosols.

## 1 Introduction

Atmospheric aerosols play a critical role in Earth's climate system; however, their effects on radiative forcing remain one of the largest uncertainties in climate models (Boucher et al., 2013; Szopa et al., 2021, Foster et al., 2021). These uncertainties are primarily due to the large heterogeneity of aerosol properties and their spatiotemporal variations. Consequently, continuous monitoring of aerosols is essential. Remote sensing instruments, such as lidars and photometers, are suitable for this purpose. Lidar provides detailed information on aerosol characterization in vertical profiles, while photometers offer integrated



columnar information. The combination of both systems allows for the continuous monitoring of the temporal evolution and
spatial distribution of aerosols, thereby reducing the uncertainties in aerosol radiative impacts by providing a more
comprehensive understanding of the optical properties of atmospheric aerosols (Granados-Muñoz et al., 2014; Bovchaliuk et
al., 2016; Tsekeri et al., 2017; Lopatin et al., 2021; Dos Santos Oliveira et al., 2023; López-Cayuela et al., 2023).
The extensive worldwide distribution of the automatic CE318 sun-sky photometer within the open-access AERONET network
(AErosol Robotic NETwork; Holben et al., 1998; Giles et al., 2019), has demonstrated its capability to provide multispectral
and multiangular information on aerosol properties (Dubovik and King, 2000; Smirnov et al., 2000; Torres et al., 2017).
Multiple studies using photometer observations have showcased the heterogeneity in aerosol properties according to their
sources and environmental conditions (Dubovik et al., 2002; Torres and Fuertes, 2021; Boichu et al., 2023). Lidar observations
are more limited due to the high cost and maintenance of the specialized systems (i.e., high-power laser sources). Efforts to
increase knowledge of vertical variations in aerosol properties have resulted in the use of ceilometers, initially manufactured
for determining cloud altitude using near-infrared (NIR) light (Cazorla et al., 2017; Jin et al., 2018; Li et al., 2021; Adam et
al., 2022; Bedoya-Velásquez et al., 2022), and micropulse lidar systems using low-power lasers (in the order of microjoules)
in the visible spectrum (Welton et al., 2001; Campbell et al., 2002; Córdoba-Jabonero et al., 2021; Barreto et al., 2022a; López-
Cayuela et al., 2022; Lopatin et al., 2024). Both types of lidar systems have demonstrated good performance in continuous
monitoring and assessing aerosol optical properties at one wavelength, especially when collocated with photometers (Mortier
et al., 2013; Popovici et al., 2018, 2022; Bedoya-Velásquez et al., 2022; Lopatin et al., 2024). Moreover, the addition of
polarization measurement capabilities has enhanced the ability to differentiate non-spherical particles, thus improving aerosol
typing (Burton et al., 2013; Groß et al., 2013; Floutsi et al., 2023; López-Cayuela et al., 2023).
As a solution for continuous monitoring of aerosol properties, the French company CIMEL have proposed the combined use
of the CE376 micropulse lidar, featuring two wavelengths and polarization capabilities, along with the CE318-T sun-sky-lunar
photometer (Barreto et al., 2016). The inclusion of a second wavelength in the lidar system provides access to information on
aerosol size. The software developments used in this study were performed in the frame of the joint laboratory between CIMEL
and the LOA (Laboratoire d'optique atmosphérique) called AGORA-Lab. Previous studies using the CE376 lidar have
presented its capabilities in atmospheric monitoring (Riandet et al., 2023; Papetta et al., 2024), and the first results using both
instruments were presented by Sanchez-Barrero et al. (2024).
This study was conducted at the Izaña Atmospheric Research Centre (IARC, 28.31° N, 16.50° W), a multiplatform site
providing long-term measurements of atmospheric components (Cuevas et al., 2024), part of the State Meteorological Agency
of Spain (AEMET). The site is located on a mountaintop at 2,367 meters above sea level (with an average pressure of 770
hPa), in the vicinity of Teide National Park in the Canary Islands. This archipelago, located in the subtropical North Atlantic,
is characterized by a very stable and well-stratified lower troposphere with strong temperature inversions (Cuevas, 1995;
Torres et al., 2002; Carrillo et al., 2016). The atmospheric stability observed in this region is modulated by the quasi-permanent
northwest subsidence regime prevailing in the free troposphere and attributed to the descending branch of the Hadley cell
around 30°N. Izaña typically resides above the temperature inversion layer, often situated between 900 and 800 hPa (Torres
et al., 2002; Carrillo et al., 2016). The location of the Izaña observatory plays a key role in studying the local and regional
transport of atmospheric aerosols in the North Atlantic. Background conditions prevail throughout the year, influenced by air
masses originating from the mid-troposphere of the North Atlantic. These events occur most frequently in April and June
(Torres et al., 2002). Conversely, dust-laden conditions are predominant during summer when the inversion is more
pronounced (Barreto et al., 2022b). The scientific programs conducted at the Izaña site encompass a comprehensive collection
of chemical and aerosol species data acquired through both in-situ measurements and remote sensing techniques.
We focus on the aerosol events that impacted the Canary Islands from August 2021 to August 2023. The motivation for this
paper is twofold. Firstly, we introduced an uninterrupted (24/7) intercomparison campaign spanning 18 days, to evaluate the
performance of a CE376 GPNP lidar (CIMEL, hereafter CE376) relative to the compact lidar MPL-4B (MPLNET) collocated
at the site. Secondly, we provide a comprehensive aerosol typing characterization using aerosol optical properties derived from
CE376 lidar profiles during the occurrence in the region of Saharan dust outbreaks, volcanic eruptions and wildfires. This
study demonstrates that using a two-wavelength micro-lidar, different aerosol types can be easily distinguished and
characterized during continuous monitoring and highlights the potential of the CE376 lidar as a tool for evaluating the vertical
distribution a temporal evolution of atmospheric aerosols.



Gaining knowledge into the distribution and characteristics of aerosol particles on both regional and global scales plays an
important role in minimizing the uncertainties of models predicting future climate scenarios. In this context, continuous
measurements from affordable polarization lidar instruments are paving the way for enhanced global monitoring of the vertical
distribution of aerosols and their properties.

**2 The CE376 lidar: specifications, calibration and retrieval methods**
**2.1 Specifications**
The CE376 is an autonomous, compact elastic backscatter lidar that operates at two distinct wavelengths: 532 nm and 808 nm.
It emits an eye-safe pulsed Nd:YAG laser light at 532 nm with an energy of 5.8 µJ, a pulse width of 0.89 ns, and a pulse
repetition rate of 4.6 kHz. This laser is directed through a set of dichroic mirrors and collimation lenses. The field of view
(FOV) for emission at 532 nm is 50 µrad, while for reception, it is 120 µrad. For the 808 nm channel, we use a pulsed laser
diode with a spectrally narrow linewidth of 0.6 nm, achieved through a volume Bragg grating (VBG). The laser operates at an
energy of 1.9 µJ and a repetition rate of 4.7 kHz. It is coupled with an optical fibre and collimation lenses. At 808 nm, the FOV
is wider, with emission and reception FOVs of 230 µrad and 260 µrad respectively. This system measures elastic backscattered
light and depolarization at both wavelengths. Its optical design consists of two Galilean telescopes in a biaxial configuration.
At each wavelength channel, the elastic backscattered light is gathered and detected using Excelitas avalanche photodiode
detectors operating in single photon counting mode (SPCM-APD) operating in photon counting mode. The dead time of these
channels varies between 23 and 28 ns. The electronic cards responsible for data control and acquisition were developed by
CIMEL.
The system has a range resolution of 15 meters, operates continuously (24/7) in 1-minute integration intervals and measures
depolarization at both 532 nm and 808 nm, thus enhancing its capacity for aerosol characterization. Linear depolarization
measurements are available by splitting the backscattered light into parallel and perpendicular components relative to the
incident plane with Polarizing Beamsplitter Cubes (PBS). In this former model, we regulate the polarization plane of the
incident light with an accuracy of 2 degrees with manual Half-Wave Plates (HWP) positioned in front of the PBS within each
channel.
The CE376 uses the Aerosol Optical Depth (AOD) data from the collocated CE318-T master photometer at Izaña station to
better constrain the Klett-Fernald solution. This photometer manufactured by CIMEL, performs direct solar and lunar
irradiance measurements at nine wavelengths (340, 380, 440, 500, 670, 870, 936, 1020, and 1640 nm; Barreto et al., 2016).
The instrument derives spectral AOD with an accuracy up to 0.01, with higher uncertainties in the UV channels. The extinction
Ånsgtröm exponent (EAE) is determined by pairs of AOD values at different wavelengths and provides information of the
aerosol size (Ångström 1929; Kusmierczyk-Michulec, 2002). Automatic near real time aerosols properties are retrieved by the
CE318-T at Izaña following the AERONET standards (AERONET; https://aeronet.gsfc.nasa.gov) (Holben et al, 1998; Giles
et al, 2019). Aerosols microphysical properties, such as volume size distribution (VSD), refractive index and single-scattering
albedo can also be retrieved from multi-angular sky radiance measurements and inversion procedures (Dubovik and King,
2000; Sinyuk et al, 2020).

**2.2 Signal processing and calibrations**
The backscattered light from molecules and aerosols at a distance r in the atmosphere is collected by the CE376 and detected
using photon counting. The signal is detected at two wavelengths, 532 and 808 nm, in two configurations: parallel (∥) and
perpendicular (X) polarized signals to the receiver (532 ∥, 532 X, 808 ∥ and 808 X). The detected elastic backscattered signal,
known to as the range-corrected signal (RCS, in Ph s$^{-1}$ m$^2$), undergoes a series of instrumental corrections for the SPCM
detector linearity, background, overlap function O(r), and range dependence (r$^2$) at each channel. The RCS equation (Eq. 1;
Weitkamp, 2005) includes information about: 1) the calibration constant C$_{L,\lambda}$ in [Ph s$^{-1}$ m$^3$ sr], which depends on the lidar
system, 2) the backscatter coefficient, β(r), that represents the light scattered back to the system, in [m$^{-1}$sr$^{-1}$]; and 3) the
atmospheric transmission losses due to the scattering and absorption of light by aerosols and molecules, T$^2$. The contributions



of aerosols and molecules are expressed with the subscripts a and m, respectively. Therefore α(r) is the extinction coefficient
in [m$^{-1}$] (Eq. 2):
$$RCS(\lambda, r) = C_{L,\lambda}[\beta_m(\lambda, r) + \beta_a(\lambda, r)]T_{mol}^2(\lambda, r)T_a^2(\lambda, r),$$ (1)
$$T_{mol}^2(\lambda, r)T_a^2(\lambda, r) = \exp\left(-2\int_0^r \alpha_{mol}(\lambda, r')dr'\right)\exp\left(-2\int_0^r \alpha_a(\lambda, r')dr'\right)$$ (2)
The second integral term in equation 2 is known as AOD. Notice that the total atmospheric column AOD measured by the
photometer is used as a reference (AOD$_{Ph}$). Using the Ångström law and with the information of AOD$_{Ph}$ at 440 nm and EAE$_{Ph}$
(440/880), we obtain the AOD$_{Ph}$ at 532 nm and 808 nm. The total signal RCS at each wavelength is defined by summing the
parallel and perpendicular components and applying the relative amplification factor V* according to the polarization
calibration (Freudenthaler et al., 2009).
We follow the standard EARLINET procedure of performing a Rayleigh fit to check the far-range accuracy and the laser
pointing alignment of the CE376 (EARLINET, https://www.earlinet.org/). The procedure consists of normalizing the RCS (λ,
r) to the attenuated molecular backscatter coefficient (β$_{mol-att}$ (λ, r) = β$_{mol}$ (λ, r)·T$_{mol}^2$ (λ, r)) in a range free of aerosols (r$_{ref}$). The
molecular properties (β$_{mol}$ (λ, r) and T$_{mol}^2$ (λ, r)) can be determined from pressure and temperature profiles from radiosondes
or as approximations from standard atmospheric models. The same procedure can be used to determine C$_{L,\lambda}$, where the
unknown term, T$_a^2$ (λ, r$_{ref}$), can be obtained from a collocated sun-moon photometer whether it is available (Cazorla et al.,
2017). The total attenuated backscatter coefficient, β$_{att}$ (λ, r), is defined as the ratio between the RCS and the lidar constant
(Eq. 3), which describes purely atmospheric parameters.
$$\beta_{att}(\lambda, r) = RCS(\lambda, r)/C_{L,\lambda} = [\beta_{mol}(\lambda, r) + \beta_a(\lambda, r)]T_{mol}^2(\lambda, r)T_a^2(\lambda, r)$$ (3)
Two other atmospheric properties are directly derived from the CE376 lidar observations, the volume linear depolarization
ratio (δ$^v$) and the attenuated colour ratio (ACR). Both, δ$^v$ and ACR, account for molecular and aerosol contributions, however,
they can be considered as a first approximation of the aerosol properties, providing valuable information on the particle
morphology.
δ$^v$ is defined as the ratio of cross-polarized to co-polarized backscattered light from the atmosphere and derived following the
methods of Freudenthaler et al. (2009). The polarization channels are calibrated using the ±45° method, leading to less than
5% uncertainty on V* for δ$^v$ values up to 0.3. Calibration coefficients determined in July 2021 were used for data from August
2021 to October 2022. Wire grid polarizers were added to the PBS cubes to reduce crosstalk in November 2022. New
calibration coefficients were determined and used to evaluate the data set collected between November 2022 and August 2023.
The ACR (Eq. 4) approximates to the colour ratio (CR) and provides insights into particle size (Omar et al., 2009; Burton et
al., 2013; Wang et al., 2020; Qi et al., 2021). The β$_{att}$ (λ,r) corrected by the two-way molecular transmittance T$_{mol}^2$ (λ, r) is
given as a first calculation to the aerosol backscatter, as done by the CALIPSO (Cloud-Aerosol Lidar and Infrared Pathfinder
Satellite Observation) algorithms (Omar et al., 2009). Under the assumption of one aerosol type is homogeneously distributed
in the atmosphere, the ACR with values between 0 and 1 indicates presence of fine to large particles, respectively.
$$ACR(r) = \frac{\beta_{att}(808,r)\, T_{mol}^{-2}(808,r)}{\beta_{att}(532,r)\, T_{mol}^{-2}(532,r)} = \frac{[\beta_{mol}(808,r) + \beta_a(808,r)]}{[\beta_{mol}(532,r) + \beta_a(532,r)]}\exp\left(-2\int_0^r[\alpha_a(808,r') - \alpha_a(532,r')]dr'\right)$$ (4)

### *Uncertainties*

The CE376 system at Izaña effectively achieves full overlap at altitudes of 2.6 km above ground level (a.g.l.) for the 532 nm
channel and at 1.9 km a.g.l. for the 808 nm channel. Below these reference points, the optical efficiency is enhanced by
correcting the lidar signal using the overlap function. During the study period, the instrument underwent optical alignment and
depolarization calibration twice – in July 2021 and November 2022. For each period, an overlap function was determined
using at least 3 profiles measured over 4 h on different aerosol-free nights (AOD< 0.03). The uncertainty associated with
these overlap functions, expressed as the standard deviation from the mean, ranges from 4% in the parallel 532 nm channel up
to 10% in the 808 nm channel in the first kilometre of altitude. These uncertainties in the overlap function are in the magnitude



of 10% proposed by Welton and Campbell (2002). Data below 400 m a.g.l. are excluded from consideration due to increased
uncertainties resulting from detector saturation at close range.

## 2.3 Derived aerosol properties

A modified two-wavelength Klett inversion is used to derive the aerosol backscatter and extinction coefficients from CE376
lidar observations (Sanchez-Barrero et al., 2024). The inversion scheme includes the well-known Klett solution (Klett, 1985;
Weitkamp, 2005) in either of its integration forms, specified by the position of the boundary conditions. The backward (far
end at an aerosol-free region $r_{ref}$) and forward (near end, close to the ground $r_0$) Klett solutions are applied according to the
range detection limits at each wavelength, assuming a constant extinction-to-backscatter ratio (i.e., lidar ratio LR). The
detection limits are determined by the signal-to-noise ratio (SNR). In this work, detection limits are calculated using SNR
=1.5. The LR is iterated until the AOD derived from the solution converges to the measured value by the collocated CE318-T
photometer. During nighttime, the Klett backward solution is applicable for both, 532 and 808 nm, wavelengths due to higher
detection limits. During the daytime, solar background light limits detection, particularly for the 808 nm channel, reducing
detection capability to below 2 km above ground level. This necessitates the use of the forward solution and constraining the
inversion with an estimated AOD for the reduced profile at 808 nm.
As a result of the inversion, we obtain profiles of aerosol backscatter and extinction at the two wavelengths. Additional
parameters are the colour ratio (CR) from the pair of backscatter profiles ($\beta_a(\lambda, r)$), and the extinction Ångström exponent
($EAE_{lid}(532/808, r)$). Both parameters provide insights into aerosol size. Furthermore, by combining the depolarization
observations with the retrieved backscatter coefficients, we derive the particle linear depolarization ratio ($\delta^p$), defined by Eq.
(5). The molecular depolarization ratio ($\delta^{mol}$) is the theoretical value according to the bandwidth of the filter in front the half-
wave plate in a CE376 system ($\delta^{mol} \sim 0.004$) and R is the backscatter ratio, R= $(\beta_a(r) + \beta_{mol}(r))/\beta_{mol}(r)$. $\delta^p$ gives insights
on the aerosol shape with low values (close to 0) indicating spherical aerosols and values above 0.2 indicating predominance
of non-spherical aerosols (Gasteiger and Freudenthaler, 2014; Floutsi et al., 2023).
$$\delta^p(r) = \frac{[1+\delta^{mol}]\,\delta^v(r)R(r)-[1+\delta^v(r)]\,\delta^{mol}}{[1+\delta^{mol}]R(r)-[1+\delta^v(r)]} \qquad (5)$$
Detailed procedures and uncertainties are addressed in Sanchez-Barrero et al. (2024). The overlap function and polarization
calibration coefficients were determined using the data analysis software developed by CIMEL Electronique, called iAAMS.

## 3 Validation of the aerosol products of the CE376

We evaluated the CE376 aerosol products against an MPL-4B micro-pulse lidar operated by the NASA Micro Pulse Lidar
Network (MPLNET) (Welton et al., 2001, Welton et al., 2018). The CE376 was installed in August 2021 at Izaña, while the
MPL-4B lidar has been measuring at the site since February 2020. The instruments were in separate buildings distancing 223
meters, within the facilities of the Izaña observatory (Figure 1). Considering the extensive usage and established reputation of
the MPLNET data, along with the MPL-4B design similarities to the CE376, we view this intercomparison campaign as an
ideal opportunity to demonstrate the capabilities of the CE376. However, in this study we emphasize that the MPL-4B used
here relates to the MPLNET deployed and managed instrument type, not those operated outside MPLNET as calibration and
processing methods may be different.
The MPL-4B operates at 532 nm with depolarization capability in a 24/7 mode. It is characterized with a low pulse energy (5-
6 µJ) laser with a repetition rate of 2.5 kHz and depolarization detection (Campbell et al., 2002; Flynn et al., 2007; Welton et
al., 2018). Unlike the CE376, the MPL-4B has a monoaxial configuration which results in a long overlap range (typically
beyond 3 km) but does simplify some aspects of the polarization calibration. MPLNET manages the MPL-4B calibrations,
signal processing, and retrieval products. The primary calibrations include detector deadtime and dark current, laser-detector
cross-talk (afterpulse), overlap, and polarization (Campbell et al., 2002, Welton et al., 2018). Relative attenuated backscatter
signals and uncertainties are processed (Welton and Campbell, 2002), and the resulting data are used to retrieve a wide variety



of variables in their product suite, with error propagation methods producing uncertainties for the retrieved variables. MPLNET
Version 3, Level 1.5 signal data and aerosol product (Welton et al., 2000; Welton et al., 2002) were used in this study.
As explained in Sanchez-Barrero et al. (2024), the uncertainties associated to each retrieved parameter in the CE376 system is
calculated using error propagation based on first-order derivatives, as described in the literature referenced therein. In the
CE376 model, the main sources of error are the estimation of the overlap function, background noise, lidar constant, and
depolarization calibrations. Our uncertainties in the overlap function varied from 4% in the parallel 532 nm channel up to 10%
in the 808 nm channel within the first kilometre of altitude, and those in the depolarization calibration were estimated to less
than 5%. In addition, relative errors > 15 % in the extinction coefficients lead to absolute uncertainties > 0.5 in EAE (Sanchez-
Barrero et al. 2024 and references therein). Both instruments use the AOD data (with 0.01 uncertainty) from the AERONET
photometer collocated at Izaña (hereafter $AOD_{Ph}$), to better constrain the Klett-Fernald solution when available. The
convergence of the AOD leads then to the uncertainties of the LR. To compare the aerosol products of both systems, the CE376
profiles collected with a vertical resolution of 15 m were averaged to the 75 m vertical resolution of the MPL.
We used the first period of collocated measurements to conduct an intercomparison campaign. For a span of 18 days (August
13-30, 2021), both the CE376 and the MPL-4B were engaged in measurements at the Izaña site. We analysed the volume and
particle linear depolarization ratios ($\delta^v$ and $\delta^p$) and attenuated backscatter ($\beta_{att}$) and extinction ($\alpha_a$) coefficient profiles measured
at 532 nm with both, the MPL-4B (version 3 products) and the CE376. An overview of the atmospheric aerosol properties
observed by the CE376 during the campaign period is shown in Figure 1a-e. Measurements of $\beta_{att}$ at 808nm, $\alpha_a$ at 532 nm, $\delta^v$
at 532nm, ACR (808-532) recorded with the CE376, as well as AOD and extinction Ångström exponent ($EAE_{Ph}$ (532/808))
from the collocated photometer at Izaña Observatory, describe the temporal and vertical evolution of the aerosol dust event.
The grade of impact of the Saharan dust over the Canary Islands on August 16, 2021, can be observed in the Modis satellite
image (https://firms.modaps.eosdis.nasa.gov) shown in Figure 1g. The additional information given by the 808 nm channel of
the CE376 is later discussed in more detail in section 4.1.





**Figure 1.** Measurements of a) attenuated backscatter ($\beta_{att}$) at 808nm, b) aerosol extinction ($\alpha_a$) at 532nm, c) volume linear depolarization ratio ($\delta^v$) at 532nm, d) attenuated colour ratio (ACR) and e) AOD at 532 and 808nm (AOD$_{Ph}$ (532) and AOD$_{Ph}$ (808)) and extinction Ångström exponent (EAE$_{Ph}$ (532/808)) from the collocated photometer taken by the CE376 during the intercomparison period with the MPL4B lidar system. The aerosol properties registered during this period describe the typical impact of the Saharan Air Layer over Izaña during summertime. f) Location of the CE376 and MPL44161 lidar systems during the comparison campaign carried out in August 2021. Reference: Google (2023) Montaña de Izaña. Imágenes ©2023 CNES / Airbus, GRAFCAN, Maxar Technologies [accessed on January 31, 2024]. g) Modis satellite image showing the impact of the Saharan dust over the Canary Islands on August 16, 2021 (https://firms.modaps.eosdis.nasa.gov). Yellow dashed bars in (a)-(e) highlight the time frames selected to calculate the averaged profiles described in Figure 2.

From August 14-18, 2021, a relatively homogeneous layer of Saharan dust aerosols from North Africa of nearly 7 km of thickness arrived to Izaña. During this period, $\delta^v$ reached values close to 0.2, AOD ~ 0.3 at 532 nm and EAE$_{Ph}$ (532/808) of 0.2-0.5 in agreement with previous studies (Figure 1c, Freudenthaler et al., 2009; Haarig et al., 2022). In the following days, August 19-21, the dust layer gradually diminished (3.5 km of thickness, EAE$_{Ph}$ (532/808) ~ 0.3) to then lead to the





characteristic maritime clean aerosol conditions at the site (August 23-25, 2021, Figure 1c). Subsequently, on August 25-26,
2021, air masses from southern regions of the African continent (South of Mauritania, Senegal and Niger) were detected at
Izaña evolving from fine, low depolarizing particles ($\delta^v$ = 0.03±0.01, $EAE_{Ph}$ (532/808) ~ 0.4, Figure 1c) to coarser and higher
depolarizing particles by August 28, 2021 ($\delta^v$ ~0.2 and $EAE_{Ph}$ (532/808) ~0.3, Figure 1c). The temporal evolution and intensity
of the Saharan dust event is also well described using the combined information given by the $\delta^v$ (532), $\alpha_a$ (532) and ACR (808-
532). The highest values accurately depict the peak of the dust episode, followed by a decline as the dust dissipates.
To estimate the overall difference between the CE376 and the MPL4B, we calculated the 15-min averaged $\delta^v$ (532) profiles
measured simultaneously by the two instruments over the entire study period. The retrievals of the CE376 were defined with
a blind zone of 400 m at 532 nm and 350 m at 808 nm, based on the overlap function. Using the whole data set, the absolute
difference in LR (532) is 9 ± 14 sr, that drops to 2 ± 13 sr when nighttime data (21h - 6h) is used. The absolute differences
between the CE376 and the MPL4B on $\delta^v$ (532) using 15-min average signals, were 0.003±0.005 (mean ± std) in the first 4
km of the profiles (6.5 km a.s.l) using the complete dataset (0-24h). Above this altitude, the difference between the two
instruments increases during daylight, mainly due to the larger effect of the solar background on the CE376 (0.01±0.06). This
effect is particularly pronounced during the summer months at the site's latitudes when the sun is near the zenith. The absolute
differences in $\delta^v$ decrease to 0.002±0.01 along the 7.5 km of the profile (10 km a.s.l.) when only the data collected during
nighttime is selected. The absolute differences on $\alpha_a$ and $\beta_a$, measured at 532 nm by both instruments, were 1.3±3.2 Mm$^{-1}$ and
0.14±0.12 Mm$^{-1}$sr$^{-1}$, respectively. This absolute difference was calculated as the difference between the 15-min average signals
and using the first 4 km of the profile (6.5 km a.s.l) and the complete dataset. The absolute differences in extinction and
backscatter are also reduced when only nighttime data is used.
In Figure 2, we selected three different time frames (T1, T2, T3) that summarize the significant variability of dust aerosol
profiles during the comparison. The aerosol properties presented in the Figure 2 (from left to right) are $\delta^v$ (532, 808), $\alpha_a$ (532,
808), $\beta_a$ (532, 808), $\delta^p$ (532, 808), EAE (532/808) from the CE376 and the collocated photometer, and CR (808-532) and ACR
(808-532). Each row represents a specific time frame based on 15-minute averages. Profiles of $\delta^v$, $\delta^p$, $\beta_a$ and $\alpha_a$, acquired
simultaneously by the CE376 (in green dots) and MPL-4B at 532 nm (blue lines), highlight the good agreement between the
two instruments, depicting the vertical aerosol structure similarly. Minor discrepancies in magnitude are mainly attributed to
errors arising from the determination of the overlap function and the depolarization calibration in each instrument.






**Figure 2. Three examples showing dust aerosol profiles catching the impact of the Saharan Air Layer on August 2021. Each row represents a 15-min profile on a particular time of a selected day and each column the different retrieved parameters. Each example includes profiles of $\delta^v$, $\alpha_a$, $\beta_a$ and particle linear depolarization ($\delta^p$), acquired simultaneously by the CE376 (green dots) and MPL-4B (blue lines) at 532 nm. The profiles measured simultaneously at 808 nm by the CE376 (red lines) are also included. Profiles of Extinction Angstrom Exponent ($EAE_{Lid}$ (532/808)), colour ratio (CR (808-532)) and attenuated colour ratio (ACR (808-532)) determined with the CE376 are also shown. The EAE from the collocated photometer ($EAE_{Ph}$ (532/808)) is also included. These examples are highlighted as yellow dashed bars in Figure 1a-e.**

## 4 Results: distribution and properties of atmospheric aerosols observed with the CE376

Continuous monitoring of aerosol vertical profiles offers valuable information about the vertical distribution, temporal evolution, size, and composition of various aerosol types observed at a site. In this section we present a detailed description of several vertical profiles recorded by the CE376 during four distinct aerosol events and we explore the insights provided using two wavelengths of the CE376 for aerosol characterization.





**4.1 Saharan dust aerosols**

We used the data collected in August 2021, as presented in the previous section, to provide additional information on the aerosol properties during the impact of Saharan dust particles at Izaña in the summertime. An extensive characterization of the climatology and aerosol properties of the summer Saharan Air Layer (SAL) has been already carried out in previous studies at the site using in-situ and remote sensing instrumentation (Rodríguez et al., 2011; Barreto et al. 2022b). The export of Saharan dust towards the Atlantic Ocean during summertime critically depends on the interannual variability of the large-scale meteorology over Northwestern Africa (Rodríguez et al., 2015). The summer SAL is a quite well-mixed dry dust aerosol layer that usually extends to 6 km height. At Izaña, this layer typically shows a maximum peak around 2.5 km a.s.l. with aerosol extinction coefficients ($\alpha_a$ (532)) > 65 Mm$^{-1}$ (Barreto et al., 2022a). While the particles transported within the summer Saharan Air Layer (SAL) are primarily soil dust emissions, traces of particulate pollutants—mainly associated with oil refineries, the fertilizer industry, and power plants in Northern and Eastern Algeria, Tunisia, and along the Atlantic coast of Morocco—have also been detected (Rodríguez et al., 2011). Due to the proximity of the measurement site to the African coast, we consider these emissions to be fresh Saharan dust emissions.

Variations on the temporal and vertical evolution of aerosol properties of the dust event from August 2021 is summarized in Figure 2:

(a) T1 (2021-08-15 07:15 to 07:30 UTC, Figure 2a): This profile represents the dust event observed at Izaña from August 14-18, 2021. The event was characterized by a highly depolarizing layer of non-spherical aerosols ($\delta^v$ (532) ~0.17, $\delta^p$ (532) ~0.29) reaching an altitude of 6.2 km a.s.l. The aerosol load of the layer was relatively high, with an AOD of 0.32 at 532 nm and 0.28 at 808nm. The effective LR retrieved at both wavelengths was 50 ± 2 sr at 532 nm and 55 ± 1 sr at 808 nm. The layer also exhibited relatively high aerosol extinction ($\alpha_a$ (532) ~90 Mm$^{-1}$ and $\alpha_a$ (808) ~80 Mm$^{-1}$) and backscatter coefficients ($\beta_a$ (532) ~1.9 Mm$^{-1}$sr$^{-1}$ and $\beta_a$ (808) ~1.6 Mm$^{-1}$sr$^{-1}$). The derived EAE$_{Lid}$ (532/808) matched that from the photometer (EAE$_{Ph}$ (532/808) = 0.22) throughout the entire layer, highlighting its homogeneity along the whole column. Additionally, ACR values of ~0.6 along the layer emphasized the presence of coarse aerosols along the well-mixed layer.

(b) T2 (2021-08-20 03:00 to 03:15 UTC, Figure 2b): This profile represents the dust event observed at Izaña from August 19-21, 2021. A lower layer of dust particles (~4.3 km a.s.l.) presented similar properties to those observed in the previous days. The profile showed slightly lower depolarization ($\delta^v$ (532) ~0.15), aerosol load (AOD (532) = 0.17 and AOD (808) =0.15), aerosol extinction ($\alpha_a$ (532) ~80 Mm$^{-1}$ and $\alpha_a$ (808) ~70 Mm$^{-1}$), aerosol backscatter ($\beta_a$ (532) ~1.2 Mm$^{-1}$sr$^{-1}$ and $\beta_a$ (808) ~0.9 Mm$^{-1}$sr$^{-1}$) and ACR (808-532, ~0.5); but similar $\delta^p$. The effective LR was higher than in the previous event, with values of 61 ± 4 sr at 532 nm and 65 ± 5 sr at 808 nm. The derived EAE$_{Lid}$, in line with that from the photometer (EAE$_{Ph}$ (532/808) = 0.26), once again highlights the homogeneity of the aerosol properties within the layer.

(c) T3 (2021-08-26 23:00 to 23:15 UTC, Figure 2c): This profile represents the arrival of a new dust episode at the site, featuring a layer between 3.5 and 4.5 km a.s.l. with a peak in aerosol properties around 4 km a.s.l. This episode originated at the same latitudes as the previous event. At its peak, the aerosol properties were similar to those observed in the previous dust episode: $\delta^v$ (532) ~0.16, $\delta^v$ (808) ~0.25, $\delta^p$ (532, 808) ~0.3, $\alpha_a$ (532) ~80 Mm$^{-1}$, and $\alpha_a$ (808) ~65 Mm$^{-1}$, $\beta_a$ (532) ~1.4 Mm$^{-1}$sr$^{-1}$ and $\beta_a$ (808) ~1.1 Mm$^{-1}$sr$^{-1}$. Both, EAE$_{Lid}$ and the EAE$_{Ph}$ (532/808), were approximately 0.43. An ACR (808-532) of ~0.5-0.6 was observed. The aerosol load along the column was relatively low (0.07 and 0.06 at 532 and 808 nm), resulting in high errors in the lidar ratios (58 ± 17 sr at 532 nm and 56 ± 16 sr at 808 nm). During the 18-days of Saharan dust intrusions studied here, we generally observe a larger depolarization effect with longer wavelengths due to the non-sphericity of large Saharan dust aerosols.

**4.2 Volcanic aerosol from Cumbre Vieja**

The eruptive period of the Cumbre Vieja Volcano lasted 85 days (La Palma, Canary Islands, Spain; Figure 3f), from September 19 to December 13, 2021. Continuous monitoring of the eruption was made possible by the collaborative efforts of scientific, private, and governmental organizations. At the beginning of the eruption, fresh volcanic emissions, including sulfuric emissions, were measured on the islands. Approximately 1.8 Tg of sulphur dioxide ($SO_2$) was injected into the troposphere at altitudes between 3 and up to 6 km (Bedoya-Velásquez et al., 2022; Milford et al., 2023). The Tropospheric Monitoring



Instrument (TROPOMI) satellite onboard the Copernicus Sentinel-5 Precursor (S-5P) satellite measured volcanic $SO_2$ total
column densities up to 20 Dobson Units (DU) (Figure 3h, 2024). The S-5P, which has been orbiting in a sun synchronous
polar orbit with an equator crossing at 13:30 local solar time since August 2019, offers high spectral covering from ultraviolet
to shortwave infrared wavelengths and a spatial resolution of $5.5 \times 3.5$ km$^2$ (Theis et al., 2017). At the Izaña Observatory,
located 140 km from the volcano (Figure 3g), $SO_2$ concentrations up to 7700 μg m$^{-3}$ were recorded (Milford et al., 2023).
Subsequent changes in atmospheric conditions led to the mixing of volcanic aerosols and desert dust in the region.
Figure 3a-e shows an overview of the atmospheric aerosol properties observed during September 23-24, 2021. Temporal
evolution of $\beta_{att}$ (808), $\alpha_a$ (532), $\delta^v$ (532), ACR (808-532) from the CE376 and AOD and EAE from the collocated photometer
during these two days, show the impact of the Cumbre Vieja volcano sulfuric plumes over the Izaña Observatory. A thin
aerosol plume around 4 km a.s.l. was observed on September 23, 2021, followed by a thicker aerosol layer descending from 6
km down to 4 km a.sl. in the following day (Figure 3a-d). These two layers show relative high concentrations ($\beta_a$ (808): 0.5-
0.8 Mm$^{-1}$sr$^{-1}$, $\alpha_a$ (532): 40-100 Mm$^{-1}$, and $AOD_{Ph}$ (532): 0.05 – 0.15) of nearly non-depolarizing ($\delta^v$ (532) < 0.05) fine aerosols
($EAE_{Ph}$ (532/808): 1.5 - 2.0 and ACR <0.4). Similarly, VSD from AERONET show higher concentration of aerosols in the
fine mode (radius of 0.15 μm) over the coarse mode (data available in the AERONET website, not shown for the sake of
brevity).







**Figure 3.** Temporal evolution of a) $\beta_{att}$ at 808nm, b) $\alpha_a$ at 532nm, c) $\delta^v$ at 532nm, d) ACR (808-532) and e) $AOD_{Ph}$ (532), $AOD_{Ph}$ (808) and $EAE_{Ph}$ (532/808) from the collocated photometer at Izaña Observatory showing the impact of the Volcanic plume on September 23-24, 2021. f) Imagen of the Cumbre Vieja eruption (Courtesy of AEMET). g) Schematic showing the relative location of the Cumbre Vieja volcano and the Izaña Observatory (adapted from Milford et al., 2023). h) Images of TROPOMI SO2 total vertical column density (in Dobson units, D.U.) from September 23 and 24, 2021. Images drawn using the VolcPlum interactive portal, developed in the frame of AERIS data centre and LOA (https://volcplume.aeris-data.fr/, Boichu and Mathurin, 2022). Yellow dashed bars in (a)-(e) highlight the time frames selected to calculate the averaged profiles described in Figure 4.



For further analysis of the aerosols observed at Izaña, we selected three-time frames (T1, T2, T3) of one to two hours average,
that improves the signal-to-noise ratio (SNR) and results in more reliable information (Figure 4). The aerosol properties
presented in Figure 4 (from left to right) are $\delta^v$ (532, 808), $\delta^p$ (532, 808), $\alpha_a$ (532 and 808), $\beta_a$ (532 and 808) EAE (532/808)
from the CE376 and the collocated photometer and CR (808-532) and ACR (808-532). Brown dashed line boxes highlight the
aerosol layers of interest. The analysis of the results is presented below:
(a) T1 (2021-09-23 17:00 to 18:00 UTC, Figure a): the effective LR retrieved at both wavelengths were $61 \pm 10$ sr at 532 nm
and $76 \pm 19$ sr at 808 nm, both affected by large errors due to low aerosol loading ($AOD_{Ph}$ (532) =0.07). The LR at 808 nm
was retrieved using an estimated AOD at 808 nm and the forward integration Klett method to reduce the uncertainty due to
the increasing background noise caused by solar radiation (Sanchez-Barrero et al., 2024). Two well-defined layers of
approximately 0.4 km width were detected at 3 km and 3.8 km asl, respectively. The lower layer presented higher extinction
($\alpha_a$ (532) ~115 $Mm^{-1}$, $\alpha_a$ (808) ~45 $Mm^{-1}$) than the layer at higher altitude (($\alpha_a$ (532) ~60 $Mm^{-1}$ and $\alpha_a$ (808) ~48 $Mm^{-1}$). $\beta_a$
peak values were ~1.4-1.9 $Mm^{-1}sr^{-1}$ at 532 nm and ~0.6-0.8 $Mm^{-1}sr^{-1}$ at 808 nm. $\delta^p$ (532) ranges between 0.01 and 0.015, and
slightly higher values at 808 nm (0.015 - 0.02) at the peaks of the layers. The derived $EAE_{Lid}$ was similar to that from the
photometer ($EAE_{Ph}$ (532/808) = 1.58), and ACR (808-532) was <0.35. ACR was lower than 0.5, with a large uncertainty (~20
%), in relation to the LR error propagation. Similar uncertainties were also found in EAE. Although the errors were significant,
both layers exhibited a predominance of fine spherical aerosols, with a higher contribution in the lower layer compared to the
higher layer.
(b) T2 (2021-09-24 08:00 to 09:00 UTC, Figure b): the effective LR retrieved at both wavelengths were $39 \pm 5$ sr at 532 nm
and $61 \pm 10$ sr at 808 nm. This set of profiles revealed 3 thin aerosol layers (~100 m width) between 3.6 and 4.2 km a.s.l., all
three with similar extinction coefficients, around ~60 $Mm^{-1}$ at 532 nm and ~40 $Mm^{-1}$at 808 nm, and similar $\delta^p$ values around
0.1. Immediately above, a wider layer between 4.3 and 5.3 km asl was detected, with $\alpha_a$ ~60 $Mm^{-1}$ at 532 nm and ~35 $Mm^{-1}$at
808 nm, $\beta_a$ ~0.8-1.5 $Mm^{-1}sr^{-1}$ at 532 nm and ~0.6 $Mm^{-1}sr^{-1}$ at 808 nm, and $\delta^p$ considerably lower (0.03). In contrast $EAE_{Lid}$,
ACR (808-532) and CR (808-532) were likely constant within the 4 layers, all showing values related to the presence of fine
aerosols. In particular, the wider layer yields the unique presence of non-ash particles, most likely sulphate aerosols.
(c) T3 (2021-09-24 12:00 to 13:00 UTC, Figure c): the effective LR retrieved at both wavelengths were $36 \pm 2$ sr at 532 nm
and $57 \pm 7$ sr at 808 nm. This set of profiles, follow the descent of the layer centred at 4.8 km asl detected in T2, now placed
between 4 and 5 km asl with higher aerosol concentration at 532 nm (extinction: 74 $Mm^{-1}$ at 532 nm and ~40 $Mm^{-1}$ at 808 nm)
and $\delta^p$ values of 0.02. A layer below 3.5 km asl was also detected with higher values of aerosol extinction (100 $Mm^{-1}$ at 532
nm and ~50 $Mm^{-1}$ at 808 nm) and backscatter coefficients ($\beta_a$ (532) ~2.1-2.9 $Mm^{-1}sr^{-1}$, $\beta_a$ (808) ~0.6-0.8 $Mm^{-1}sr^{-1}$); and $\delta^p$
below 0.04. Both layers showed high values of $EAE_{Lid}$ (1.4 and 2, respectively). The ACR (808-532) remained similar in the
three cases shown (~0.3 – 0.4). Thus, an increasing presence of non-ash particles was evidenced, in accordance with the
increase of $EAE_{Ph}$ (1.8) with respect to T1and T2 (1.7 and 1.4, respectively). Similar to dust aerosols, we generally observe
slightly higher depolarization values at longer wavelengths.
The analysis presented here indicates the presence of likely non-ash particles (fine and spherical aerosols) at Izaña. This slightly
contrast with the more mixed presence of ash and non-ash particles observed when the aerosol profile was measured at 10 km
from the volcano on the island of La Palma. There, the lowest and strongest peak of the volcanic plume was observed just
above the top of the marine boundary layer (1.3 – 1.6 km above sea level, according to Sicard et al., 2022; Córdoba-Jabonero
et al., 2023). At the start of the volcanic eruption, ashes and particulate matter deposition was primarily observed within the
marine boundary layer, close to the emission source. Meanwhile, the Izaña station monitored the regional transport at higher
altitudes (up to 5 km a.s.l.) of a more dominant fine sulphate particle layer, as these particles can remain suspended for longer
periods and travel long distances (Graf et al., 1997).






**Figure 4. Impact of the Cumbre Vieja volcanic plume in the aerosol properties measured at Izaña expressed as average aerosol profiles measured during September 23-24, 2021. Each row represents a 1 h or 2 h profiles including profiles of $\delta^v$ (532, 808), $\delta^p$ (532, 808), $\alpha_a$ (532 and 808), $\beta_a$ (532 and 808) EAE (532/808) from the CE376 and the collocated photometer (EAE$_{Lid}$ and EAE$_{Ph}$, respectively) and CR (808-532) and ACR (808-532). The selected time frames shown here are highlighted as yellow dashed bars in Figure 3a-e.**

### 4.3 Fresh and aged aerosols from wildfires

#### 4.3.1 Aged aerosols from the Canadian wildfires

Starting in March 2023, Canada suffered from the record-breaking wildfires affecting 15 million hectares of land and with a total emission of carbon of 647 TgC by September 2023 (Byrne et al., 2024). Smoke plumes travelled all over the Atlantic, reaching the Canary Islands within approximately 10 days during May and June 2023. Examples on how these smoke plumes





travelled over the Atlantic and impacted the Izaña Observatory over these months are shown in Figure 5a-e and Figure 6a-e.
The figures include measurements of $\beta_a$ (808), $\alpha_a$ (532), $\delta^v$ (532), ACR (808-532), as well as AOD and $EAE_{Ph}$ from the
collocated photometer recorded in May 20-21 and July 01-02, 2023. These days were selected to avoid or minimize the
simultaneous impact of Saharan dust particles.
During May 20 and 21, 2023, a plume of aged biomass burning aerosols from Canada was descending from 9 km to 3.5 km
a.s.l. and detected at Izaña. The AERONET VSD indicated a single fine mode with an effective radius centred at 0.26 μm (data
available in the AERONET website, not shown for brevity), which is slightly higher than the background value of 0.16 μm
typical observed at Izaña (Barreto et al., 2022b), and in the order of magnitude of those observed in particles that have
undergone aging (Eck et al., 2009; González et al., 2020). As the plume descended to the observatory, its extinction and
backscatter properties diminished, while the values of $\delta^v$ (532) and ACR (808-532) remained relatively constant (~0.08 and
~0.05, respectively; Figure 5a-d). By the evening of May 21, 2023, the plume reached the observatory level. The lidar ratio
(LR) values were $73 \pm 5$ sr at 532 nm and $97 \pm 6$ sr at 808 nm. The aerosol properties measured at this level were $\delta^v$ (532)
~0.03, $\delta^p$ (532) ~0.06, $\alpha_a$ <120 Mm$^{-1}$ at 532 nm and <70 Mm$^{-1}$ at 808 nm, $\beta_a$<1.5 Mm$^{-1}$sr$^{-1}$ at 532 nm and <0.6 Mm$^{-1}$sr$^{-1}$ at 808
nm, ACR (808/532) ~0.5, and the $EAE_{Lid}$ given by the lidar system ~ 1.5, similar to that given by the photometer (1.43, Figure
5h).







**Figure 5. Temporal evolution of a) $\beta_{att}$ at 808nm, b) $\alpha_a$ at 532nm, c) $\delta^v$ at 532nm, d) ACR and e) $AOD_{Ph}$ (532), $AOD_{Ph}$ (808) and $EAE_{Ph}$ (532/808) from the collocated photometer at Izaña Observatory showing the impact of the long ranged transported (aged) smoke from Canadian wildfires on May 20 and 21, 2023. f) Satellite images highlighting the Canadian fires (https://firms.modaps.eosdis.nasa.gov) [accessed on June 09, 2024]. g) HYSPLIT isentropic backtrajectories showing the Canadian origin of the airmasses approximately 10 days before the arrival to the Izaña Observatory in the evening on May 20, 2023. h) 1h average profiles of $\delta^v$ (532, 808), $\delta^p$ (532, 808), $\alpha_a$ (532 and 808), $\beta_a$ (532 and 808) EAE (532/808) from the CE376 and the collocated photometer ($EAE_{Lid}$ and $EAE_{Ph}$, respectively) and CR (808-532) and ACR (808-532). The selected time frame selected to calculate the averaged profiles in (h) is highlighted as yellow dashed bars in plots (a)-(e).**





452

453

**Figure 6. Temporal evolution of a) $\beta_{att}$ at 808nm, b) $\alpha_a$ at 532nm, c) $\delta^v$ at 532nm, d) ACR (808-532) and e) $AOD_{Ph}$ (532), $AOD_{Ph}$ (808) and $EAE_{Ph}$ (532/808) from the collocated photometer at Izaña Observatory, showing the impact of the long ranged transported (aged) smoke from Canadian wildfires on July 1-2, 2023. f) Satellite images highlighting the Canadian fires (https://firms.modaps.eosdis.nasa.gov) [accessed on June 09, 2024]. g) HYSPLIT isentropic backtrajectories showing the Canadian**



**origin of the airmasses approximately 10 days before the arrival to the Izaña Observatory in the evening of July 1, 2023. h) 3-h average profiles of $\delta^v$ (532, 808), $\delta^p$ (532, 808), $\alpha_a$ (532 and 808), $\beta_a$ (532 and 808) EAE (532/808) from the CE376 and the collocated photometer (EAE$_{Lid}$ and EAE$_{Ph}$, respectively) and CR (808-532) and ACR (808-532). The selected time frame selected to calculate the averaged profiles in (h) is highlighted as yellow dashed bars in plots (a)-(e).**

A new episode of transported smoke was observed by the end of the day of July 1, 2023. This episode presented a minimal influence from the commonly observed summer Saharan dust in comparison to the previous and following days (back trajectories not shown for brevity, Figure 6a-d). The event was characterized by a quite homogenous layer of aerosols reaching 3.5 km height over Izaña. This layer is characterized by an extinction coefficient of ~149 Mm$^{-1}$ at 532 nm and ~95 Mm$^{-1}$ at 808 nm, backscatter coefficient of ~3 Mm$^{-1}$sr$^{-1}$ at 532 nm and ~1.5 Mm$^{-1}$sr$^{-1}$ at 808 nm, ACR (808-532) of 0.4 and $\delta^p$ of 0.08 at 532 nm. The EAE$_{Lid}$ (532/808) of 1.14 was slightly higher than that from the photometer. In this event, the AERONET VSD distributions exhibit a bimodal pattern, with a dominant fine mode centred also around 0.26 μm but higher in concentration relative to May, and a coarse mode with lower concentration centered at 1.7 μm (data available in the AERONET website, not shown for brevity). The results shown here, combined with lower effective LR (51 ± 3 sr at 532 nm and 62 ± 6 sr at 808 nm) compared to the example shown in Figure 5, suggest an observation of slightly increased of the aged smoke particles, mixed with the remaining contribution of dust aerosols observed the previous and posterior days.

The values of $\delta^p$ and LR at 532 nm obtained in this section fall within the range of those reported in the literature (Gross et al., 2013; Ortiz-Amezcua et al., 2017 and references therein) for Canadian aged Biomass Burning over Europe. The second example presented here shows slightly lower LR values. Although these lower values are within the expected range, they also indicate a slight mixing with the Saharan dust aerosols, which is more noticeable at longer wavelengths (808nm). In general, the episodes of Canadian Wildfires smoke show a lower depolarizing effect ($\delta^v$ and $\delta^p$ at 532 and 808 nm <0.1) and higher EAE due to a more homogenous morphology of the particles in comparison with dust aerosols.

### 4.3.2 Fresh aerosols from forest fires at Tenerife

On August 15, 2023, the island of Tenerife suffered from one of the most devastating fires in recent memory. It began at mid-altitudes and affected over 12,000 hectares largely within the forest. Five days later, on the afternoon of August 20, the fire reached the Izaña mountain, where the Izaña Observatory is located (Figure 7h-i). Thanks to the rapid response of firefighting efforts, the observatory's infrastructure was saved. Most of the measurement programs remained operational, recording the historical anomalies during this period.

**Figure 7. Temporal evolution of a) $\beta_{att}$ at 808nm, b) $\alpha_a$ at 532nm, c) $\delta^v$ at 532nm, d) ACR (808-532) and e) AOD$_{Ph}$ (532), AOD$_{Ph}$ (808) and EAE$_{Ph}$ (532/808) from the collocated photometer at Izaña Observatory showing the impact of the wildfires from the forestal park in Tenerife in August 17-19, 2023. f) Black carbon (BC) concentrations measured at 1 micron size cut (PM$_1$) with a MAAP (Thermo™ at 637 nm) at Izaña. g) Particle matter concentrations measured at 2.5- and 10-microns size cut (PM$_{2.5}$ and PM$_{10}$) with a 1405-DF (Thermo Fisher Scientific) at Izaña. h) Satellite images highlighting the wildfire at Tenerife and the dispersion plume (https://firms.modaps.eosdis.nasa.gov) [accessed on June 13, 2024]. i) Image of the Teide National Park taken on 19 August from Izaña Observatory (Teide Cloud Laboratory Project, #TeideLab, Courtesy of AEMET). Yellow dashed bars in (a)-(e) highlight the time frames selected to calculate the averaged profiles described in Figure 8.**

Measurements of $\beta_{att}$ at 808nm, $\alpha_a$ at 532 nm, $\delta^v$ at 532nm, ACR (808-532) from the CE376, and the AOD and EAE$_{Ph}$ from the collocated photometer, taken at Izaña during August 17-19, 2023, show the impact of the local wildfires (Figure 7a-e). On



the evening of August 18, 2023, and the following days, the trajectories of the air masses and satellite images indicated the
presence of Saharan dust particles and cloudy conditions at Izaña. In addition, when the fire was surrounding the observatory
on the evening of August 20, 2023, the virulence of the fire was translated into saturated signals of the lidar and photometer
systems, making impossible the retrieval of aerosols properties at this stage.
In the previously shown study cases, we used AERONET AOD Level 2.0 to retrieve the aerosol properties. However, in this
case, we used AOD level 1.0 because, due to the intensity of the event, most of the photometric measurements were incorrectly
screened by AERONET control algorithms and attributed to the presence of clouds, as happened during the desert dust outbreak
in February 2020 (Cuevas et at., 2021).To ensure the retrieval of aerosol properties under clear sky conditions, we used 1-
minute collocated measurements of global and diffuse short-wave downward radiation (García et al., 2019), following the
methodology proposed by Long and Ackerman (2000) and adapted by García et al., (2014) to suit the specific conditions at
Izaña. Increasing values of $\beta_{att}$ (808) and $\alpha_a$ (532), and lower $\delta^v$ (532) and ACR (808-532), with AE close to 2 were especially
observed around the afternoon, particularly on August 17 and 18, 2023. This phenomenon was due to the upslope winds, active
during daylight, carrying upwards the burned material from the lower forest (1300-1500 m). On August 19, 2023, with the fire
getting closer to the observatory and with the influence of Saharan dust conditions (Figure 7h-i), the measured aerosol showed
lower values of $\beta_{att}$ (808) and $\alpha_a$ (532), higher $\delta^v$ (at 532, >0.25) and $EAE_{Ph}$ dropped close to 0.
We selected four time frames (T1-T4; Figure 8) covering the different aerosol scenarios shown in Figure 7 and we analysed
the aerosol properties of this event using average profiles of $\delta^v$ (532, 808), $\delta^p$ (532, 808), $\alpha_a$ (532 and 808), $\beta_a$ (532 and 808),
EAE (532/808) from the CE376 and the collocated photometer, and CR (808-532) and ACR (808-532). The results are
presented below:
(a) T1 (2023-08-17 08:00 to 09:30 UTC, Figure 8a): the effective LR retrieved were 53 ± 2 sr at 532 nm and 71 ± 2 sr at 808
nm for AOD of 0.15 and 0.11, respectively. During the early hours as well as in the evening, a quite homogeneous layer up to
4.5 km height was observed. This layer presented $\alpha_a$ values of ~85 Mm$^{-1}$ at 532 nm and ~75 Mm$^{-1}$ at 808 nm, $\beta_a$ of ~1.5 Mm$^{-1}$sr$^{-1}$ and ~1 Mm$^{-1}$sr$^{-1}$ at 532 nm and 808 nm, and $\delta^p$ (532) of 0.35. $EAE_{Lid}$ (532/808) of ~0.4, like that from the photometer,
and ACR values ~ 0.35, associated with fine aerosols were observed. This aerosol layer defined the background aerosol levels
of the wildfire measured at Izaña and originated at 1000-1500 m lower than the station.
(b) T2 (2023-08-17 14:45 to 15:00 UTC, Figure 8b): we observed an increase in AOD, EAE (532/808) ~1.3 and $\alpha_a$ (~150 Mm$^{-1}$ and ~85 Mm$^{-1}$ at 532 and 808 nm), and $\beta_a$ (~3.7 Mm$^{-1}$sr$^{-1}$ and ~2 Mm$^{-1}$sr$^{-1}$ at 532 and 808 nm) coefficients, along with a lower
depolarization ($\delta^p$ (532) ~0.15), with the activation of the upslope flow winds during daylight hours. LR were lower, 42 ± 2 sr
at 532 nm and 45 ± 2 sr at 808 nm. This indicated the arrival of freshly burned material at the station, carried upward by the
wind.
(c) T3 (2023-08-18 08:00 to 09:30 UTC, Figure 8c): With the evolution of the days, the wildfire extended to higher altitudes,
getting closer to the station. In the morning of August 18, we observed a thicker layer, reaching 5.5 km, characterized by LR
of 51 ± 3 sr at 532 nm and 65 ± 3 sr at 808 nm, and AOD of 0.23 and 0.17 at 532 nm and 808 nm, respectively. It also showed
higher $\delta^p$ (532) and lower $\beta_a$ and $\alpha_a$ signals relative to the previous morning. The layer was characterized by an $EAE_{Lid}$
(532/808) of ~0.75 close to that given by the photometer. A similar impact of the upslope winds bringing freshly burned
material to the station was also detected that day.
(d) T4 (2023-08-19 15:00 to 17:00 UTC, Figure 8d): On the 19$^{th}$, the coupling of two aerosol layers, reaching a total height of
6 km, was observed. These layers were characterized by higher AOD and depolarization values ($\delta^p$ (532) >0.3), compared to
the previous days. Additionally, the second layer presented higher $\beta_a$ and $\alpha_a$ signals around 4.5 km. The effective LR of 59 ±
2 sr at 532 nm and 86 ± 2 sr at 808 nm for AOD of 0.32 and 0.29, respectively. These observations were attributed to the
combined effects of advancing fire towards the observatory and the arrival of Saharan dust particles (air masses not shown for
brevity).
During the FIREX-AQ campaign, a similar lidar system prepared for mobile applications, showed comparable ranges of values
for LR, $\delta^p$ (532), EAE (532/808), and ACR (808-532). However, it recorded higher extinction and backscatter coefficients,
which could be related to their measurement of aerosols predominantly in the fine mode and the intensity of the wildfire and
fuel available (Sánchez-Barrero et al., 2024). On the other hand, the extinction, backscatter levels and LR found here are in
the range of those measured by Alados-Arboledas et al. (2011) in fresh biomass burning aerosols. The examples presented



here showed overall quite homogeneous layers of $\delta^v$ (532, 808) between 0.1-0.2 and ACR (808-532) between 0.3-0.5. These results suggest well-mixed layers of fine and coarse smoke aerosols of varying sizes and shapes from the forested area. This is consistent with the ash, soot, and charred vegetation that surrounded the observatory during the event, as measured by the in-situ instrumentation at Izaña Observatory. During the wildfire, black carbon concentrations (BC) measured with a $PM_1$ size cut (1 micron) reached the highest levels ever recorded at Izaña, 16,000 ng·m$^{-3}$ (Figure 7f, MAAP instrument - Thermo™ at 637 nm, background levels <100 ng·m$^{-3}$). The particle matter concentrations at 2.5- and 10-microns size cut ($PM_{2.5}$ and $PM_{10}$) measured with a TEOM 1405-DF (Thermo Fisher Scientific), reached values of 600 and 700 μg·m$^3$, respectively (Figure 7g). A closer range between $PM_{2.5}$ and $PM_{10}$ concentrations during August 17 and 18, and more distancing values on the 19$^{th}$, also described the transition between the observation of more pure wildfire aerosol emissions in the first two days and the mixture with dust aerosols by August 19, 2023 (Figure 7).



**Figure 8.** Average aerosol profiles of $\delta^v$ (532, 808), $\delta^p$ (532, 808), $\alpha_a$ (532 and 808), $\beta_a$ (532 and 808), EAE (532/808) from the CE376 and the collocated photometer (EAE$_{Lid}$ and EAE$_{Ph}$, respectively) and CR (808-532) and ACR (808-532) showing the impact of the local wildfires in the aerosol properties measured at Izaña during August 17-19, 2023. The selected time frames shown here are highlighted as yellow dashed bars in the in Figure 7a-e.





562

## 5 Aerosol classification.

A summary of the retrieved aerosol properties of the events studied in the previous section is shown in Table 1. Fresh Saharan dust particles exhibited the lowest $EAE_{Lid}$ (532/808) (0.30±0.16) and relatively high $\delta^p$ (532) values of 0.20±0.01, indicating that they were the largest particles in this study and had non-spherical morphology. Similar values of LR (532) and $\delta^v$ (532) have been reported in literature (Sanchez-Barrero et al., 2024; Barreto et al., 2022b; Gross et al., 2013; Tesche et al., 2011; Tesche et al., 2009a; Tesche et al., 2009b) as well as for $\delta^p$ (532) (Haarig, et al., 2022; Gross et al., 2011; Gross et al., 2013). Higher $\delta^v$ values at 808 nm relative to those at 532 nm ($\delta^v$ (808) = 0.25±0.02), highlight the greater depolarizing and more efficient scattering effect of large non-spherical fresh dust particles at higher wavelengths. Dust also exhibited the highest ACR (808-532) values of 0.54±0.08 compared to the other studied emission sources, consistent with previous reported results (Sanchez-Barrero et al., 2024; Comeron et al., 2017).

Fresh emissions from the local forest fires on the island showed a slightly higher EAE (532/808) (0.68±0.53), lower ACR (808-532) (0.40±0.07) and higher $\delta^p$ (532) (0.30±0.11). Comparable values using a similar lidar system were observed during the FIREX-AQ campaign (Sánchez-Barrero et al., 2024). Extinction and backscatter coefficients and LR measured here at 532 nm, are also in the range of those from fresh biomass burning aerosols shown by Alados-Arboledas et al. (2011). As discussed in section 4.3.2, these aerosols appeared to be a homogeneous mixture of different aerosol types and sizes, consistent with the presence of ash, soot, and charred vegetation in the observatory, which explain the higher particle depolarization ratio for the mixture.

Long-range transported (aged) aerosols from the Canadian biomass burning, observed at the station after a 10-day journey across the Atlantic Ocean, presented values of EAE (532/808) between 1 and 1.3, and relatively low depolarization ($\delta^p$ (532) = 0.08±0.01, $\delta^p$ (808) = 0.02±0.01). These values describe small particles with a quite homogeneous (spherical) morphology and agree with those previously reported during the observations of Canadian aged Biomass Burning aerosols over Europe (Gross et al., 2013; Ortiz-Amezcua et al., 2017 and references therein). The ACR (808-532) in these events were similar to those measured during the fresh local forest fires (0.44±0.01). We also observed a slight increase in the fine mode and the emerge of a second mode centred at 1.7 um in the second event shown, suggesting aerosol ageing along the month as suggested by previous studies (Eck et al., 2009; González et al., 2020) together with the mix of trace contributions of dust aerosols observed around the event. The influence of dust is observed in the 808 nm channel, with $\delta^p$ increasing from 0.02 to 0.05.

Finally, $\delta^p$ (532, 808) of 0.03±0.01, EAE (532/808) of 1.48±0.02, and an ACR (808-532) of 0.3±0.02, describe the small, non-depolarizing sulphate aerosols from the initial days of the Cumbre Vieja volcano eruption. While ashes and particulate matter deposition was primarily observed within the marine boundary layer (Sicard et al., 2022; Córdoba-Jabonero et al., 2023) in the surroundings of the volcano, Izaña monitored the typical regional transport of a more dominant fine sulphate particle layer (Graf et al., 1997).

We observe a good agreement between the EAE shown here and those compiled by Floutsi et al. (2023). As pointed by the results and shown in previous studies (Gross et al., 2013), particle linear depolarization ratio is a key measurement for aerosol classification, well differentiating different aerosol types. In contrast, better constrained lidar ratios using sun photometers, alone, still not good for aerosol typing when using alone. This is also the case for aerosol classification using aerosol extinction or attenuated backscatter coefficients. It is important to highlight the smaller range of values of the encountered ACR and the similar values obtained for aged and fresh wildfire emissions, but distinguishable from other aerosols sources (Figure 9a, c-d). In addition, longer wavelengths (such as 808 nm in this case) appear to be more efficient of distinguishing aerosol types and mixtures, particularly when Saharan dust particles are involved due to their heterogeneous morphology (Figure 9c-d). Using a two-wavelength elastic lidar system, we can combine direct measurements, such as ACR (808-532), with retrieved properties such as $\delta^p$ or EAE, to obtain information of the size and composition of the aerosols observed in the four aerosol types studied in this work. The results are consistent with those obtained by combining two retrieved properties such as $\delta^p$ or EAE (532/808).

**Table 1. Variability of the aerosol properties observed in the four aerosol types studied in this work. The table includes mean and standard deviation of the following parameters: LR, EAE (532/808), $\delta^p$ (532, 808), $\delta^v$ (532, 808), $\beta_a$ (532, 808), $\alpha_a$ (532, 808) and ACR**



**(808-532). The "aged Canadian wildfires" corresponds to the event of May 2023. "Aged Canadian wildfires + dust" corresponds to**
**the mixing event observed in June 2023 that also received a small influence of Saharan dust particles.**

| Events<br>Aerosol retrieval | Fresh Saharan dust | Fresh volcanic emissions | Fresh forest fires | Aged Canadian wildfires | Aged Canadian wildfires + dust |
|---|---|---|---|---|---|
| LR (532) (sr) | 58±8 | 40.7±5.7 | 51.3±7.04 | 73±5 | 51±3 |
| EAE (532/808) | 0.210±0.04 | 1.50±0.28 | 0.69±0.53 | 1.43±0.11 | 1.14±0.05 |
| $\delta^p$ (532) | 0.30±0.01 | 0.04±0.02 | 0.29±0.11 | 0.08±0.01 | 0.08±0.01 |
| $\delta^v$ (532) | 0.15±0.03 | 0.02±0.01 | 0.16±0.04 | 0.04±0.01 | 0.06±0.01 |
| $\delta^p$ (808) | 0.31±0.01 | 0.10±0.06 | 0.19±0.07 | 0.02±0.01 | 0.05±0.01 |
| $\delta^v$ (808) | 0.25±0.02 | 0.08±0.02 | 0.15±0.05 | 0.02±0.01 | 0.05±0.01 |
| $\beta_a$ (532) (Mm$^{-1}$sr$^{-1}$) | 1.78±0.31 | 2.01±0.48 | 2.62±0.51 | 1.41±0.27 | 2.96±0.32 |
| $\beta_a$ (808) (Mm$^{-1}$sr$^{-1}$) | 1.01±0.39 | 0.64±0.14 | 1.09±0.39 | 0.51±0.11 | 1.426±0.09 |
| $\alpha_a$ (532) (Mm$^{-1}$) | 70.33±22.04 | 58.46±22.1 | 101.41±36.70 | 67.61±17.46 | 152.41±11.97 |
| $\alpha_a$ (808) (Mm$^{-1}$) | 62.84±23.09 | 27.87±9.41 | 74.54±19.92 | 37.44±10.29 | 94.45±5.82 |
| ACR (808-532) | 0.54±0.08 | 0.48±0.28 | 0.40±0.07 | 0.35±0.02 | 0.43±0.01 |


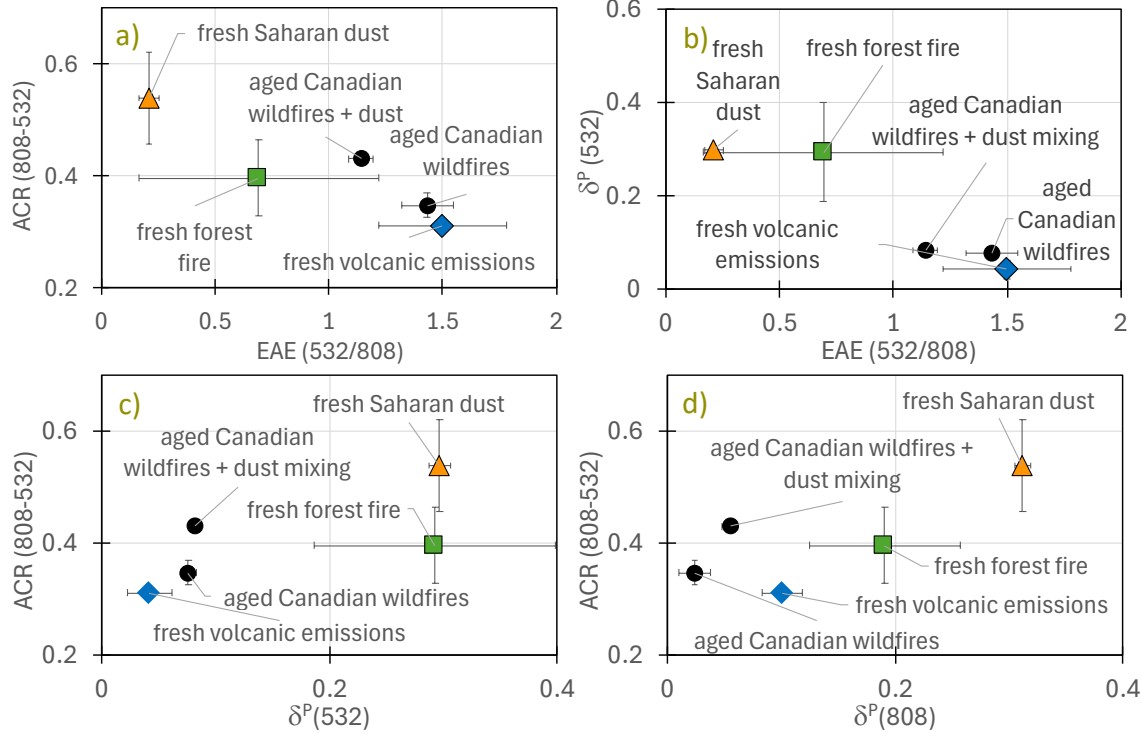

**Figure 9.** Variability of $\delta^p$ (532, 808), ACR (808-532) and EAE (532/808) observed across the different aerosol types studied in this work:
1) fresh Saharan dust particles (orange triangle), 2) fresh emission of volcanic sulphate aerosols (blue rhomboid), 3) aged smoke from
Canadian wildfires (black dots) and 4) fresh emissions from local forest fires (green square). Figures a) through d) illustrate the following
relationships: a) EAE (532/808) vs. ACR (808-532), b) EAE (532/808) vs. $\delta^p$ (532), c) $\delta^p$ (532) vs. ACR (808-532), and d) $\delta^p$ (808) vs. ACR
616 (808-532).




## 6 Conclusions

A two-wavelength CE376 micro-lidar (CIMEL) located at the facilities of the Izaña Atmospheric Research Centre (Canary Islands, Spain) was used to provide a comprehensive aerosol typing characterization of the recent aerosol events impacting this subtropical North Atlantic region.

We assessed the performance of the CE376 lidar by comparing its retrieved aerosol products with those measured by an MPL-4B lidar (MPLNET) collocated at Izaña. In general, both systems similarly reproduced the vertical aerosol structure. The main absolute differences were related to errors arising from the determination of the overlap function and the depolarization calibration of each instrument and the larger effect of the solar background on the CE376 system, especially during the central hours of the day, when the analysis is limited to the first 2 km of the atmosphere in the 808 nm channels. By studying the absolute differences in 15-minute profiles, we observed a bias of 0.3% in $\delta^v$. The absolute differences on $\alpha_a$ and $\beta_a$, measured at 532nm, were $0.1\pm0.3$ Mm$^{-1}$ and $0.14\pm0.12$ Mm$^{-1}$sr$^{-1}$, respectively, using the first 4 km of the profile (6.5 km a.s.l) and the complete dataset (0-24h). The absolute differences reduced when only nighttime data was considered, with results extended to 10 km a.g.l. The intercomparison with the MPL-4B is presented primarily to highlight the capabilities of an equivalent elastic micro-pulse lidar and to demonstrate the added value of the CE376, which offers two wavelengths and depolarization for aerosol studies. Both the CE376 and MPL lidars are earlier versions of current micro-pulse lidars. An improved design of the CE376 lidar from Cimel is now available, and the mini-MPL biaxial system has become the standard instrument within the MPLNET network. A dedicated intercomparison campaign between the latest models of both the CE376 and MPL is planned for the future.

We evaluated the vertical distribution and temporal evolution of different types of aerosols observed at the site, using the additional information provided by the two channels (532, 808 nm) of the CE376. While this mountaintop site is usually known for its pristine, dust-free conditions, it recently experienced the disruptions of three distinct types of aerosol events. Firstly, Saharan dust outbreaks regularly pass over the site towards the Atlantic especially during the summer months. Secondly, the eruption of the Cumbre Vieja Volcano (La Palma, Canary Islands) in September 2021. Lastly, a devastating wildfire occurred in Tenerife in August 2023. Additionally, aerosols transported over long distances from the Canadian wildfire plumes in May and June 2023 were also identified at the site. We showed that combined measurements of particle linear depolarization ($\delta^p$), extinction Angstrom exponent (EAE (532/808)) and attenuated colour ratio (ACR (808-532)), described the size and composition of the aerosols shown in this work. Fresh Saharan dust particles exhibited the lowest EAE (532/808) ($0.30\pm0.16$) and the highest values of ACR (808-532), together with $\delta^p$ (532) values $0.20\pm0.01$, being the largest particles in this study with non-spherical morphology. Fresh emissions from the local forest fires on the island showed a slightly higher EAE (532/808) ($0.68\pm0.53$), lower ACR (808-532) ($0.40\pm0.07$) and higher $\delta^p$ (532) ($0.30\pm0.11$), which support the mixture of different aerosol sizes and types, consistent with the presence of ash, soot, and charred vegetation in the site. Long-range transported aerosols from the Canadian biomass burning exhibited larger EAE (532/808) (1-1.3) and relatively low depolarization ($\delta^p$ (532) of $0.08\pm0.01$), indicating smaller particles with a more spherical morphology. Finally, $\delta^p$ (532) of $0.03\pm0.02$, EAE (532/808) of $1.48\pm0.02$, and ACR (808-532) of $0.3\pm0.02$, describe the small, non-depolarizing sulphate aerosols from the initial days of the Cumbre Vieja volcano eruption. These results highlight the potential of two-wavelength micro-pulse lidar systems for continuous monitoring of the temporal evolution and vertical distribution of aerosols, as well as distinguishing different aerosol types using the combined information of particle linear depolarization ratios and attenuated colour ratio.

Upcoming developments on the CE376 CIMEL micro-lidar aim to upgrade both lasers, the visible laser (532 nm) to a more powerful one, and the near-infrared laser (808nm) to reduce its sensitivity to solar background during daylight. This improvement will enhance the vertical resolution and capabilities of this system operating with a combination of two lasers at different wavelengths. As a result, it will provide more precise information on the size, composition, vertical distribution, and temporal evolution of atmospheric aerosols. Furthermore, when combined with photometric measurements, the enhanced system will offer a valuable opportunity to obtain both columnar and vertically resolved bimodal aerosol microphysical and optical properties through the Generalized Retrieval of Aerosol and Surface Properties (GRASP) inversion algorithm. This makes low-cost, dual-wavelength compact lidar systems capable of improving aerosol retrievals, both during the day and night, by leveraging their sensitivity to aerosol shape and vertical distribution. The application of GRASP is currently under study, with findings to be shared in future publications. These advancements indicate that such systems, operating 24/7 from the



ground, are powerful tools for enhancing radiative transfer models and global atmospheric models, ultimately improving predictions of the impact of atmospheric aerosols on climate.

*Data availability.* The CE376 data as well as the MPL-4B data are available on request from the Izaña WMO-Measurement Lead Centre for aerosols and water vapor remote sensing instruments (MLC). Data from AERONET used in the present study can be obtained from https://aeronet.gsfc.nasa.gov (accessed on May 05, 2024). Data from MPLNET used in the present study can be obtained from https://mplnet.gsfc.nasa.gov/download_tool/ (accessed on Feb 21, 2024). The vertical soundings can be downloaded from http://weather.uwyo.edu/upperair/sounding.html (accessed on March 14, 2024).

*Author Contribution.* YG did the data analysis and wrote and revised the paper. MFSB actively contributed to the design of the study and data analysis. EJW provided calibrated data of the MPL lidar and contributed to the writing. IP, SV and FAA actively participated in the discussion on the data analysis and calibration. RDG analysed the collocated global and diffuse short-wave downward radiation data to evaluate the presence of clouds and confirm the use of level 1 AERONET data. AB, PGS, CT and PG contributed to the writing and the paper enrichment. All coauthors provided comments on the paper.

*Competing interests.* The authors declare no conflict of interest.

*Acknowledgements.* This work is part of the activities of the WMO-Measurement Lead Centre for aerosols and water vapour remote sensing instruments (MLC). Software developments for data analysis of the CE376 lidar have been performed in the frame the AGORA-Lab initiative (https://www.agora-lab.fr). The AERONET sun photometers at Izaña were calibrated through the AEROSPAIN Central Facility (https://aerospain.aemet.es/, accessed on May 05, 2024). Action under the ACTRIS grant (agreement no. 871115). We gratefully acknowledge the data provided by the AERONET and MPLNet networks. The MPLNET project is funded by the NASA Radiation Sciences Program and Earth Observing System.

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
