# Peer review of "Compact dual-wavelength depolarization lidar for aerosol characterization over the Subtropical North Atlantic"

_EGUsphere, 2024_

## Author Response (AR1)

**RESPONSE TO REFEREES**

17 February 2025

We thank the reviewers for their time and for providing positive feedback and constructive comments on manuscript egusphere-2024-2727. We have revised the manuscript accordingly and incorporated all the proposed changes. The changes made are tracked in the document "egusphere-2024-2727-ATC2.pdf". A cleaned version of the manuscript is also attached as "egusphere-2024-2727-manuscript-version3.pdf".

In the following lines we address the reviewers concerns, following the order of the Review Report.

Sincerely,

Yenny González on behalf of coauthors

**Reviewer 1**

*"The authors present a comparison study of two lidar instruments (one well established and one rather novel, upgraded one) together with an interesting, however not highly novel, set of measurements and products (optical properties from the rather novel instrument) which can be incorporated in aerosol-typing, -monitoring, and -modelling efforts. Therefore, the manuscript is suitable for publication in AMT and I suggest the publication after minor revisions after addressing the points listed below"*.

**Major comments**

*Comment # 1:* Line 1–2: I suggest modifying the title of the publication slightly. First, I would avoid capitalization. Second, I would also avoid "North Atlantic aerosol" but write "different aerosol types observed at the North Atlantic" or similar.

*Reply*: We agree with the reviewer and have removed the capitalization and modified the title to make it more appropriate. The new tittle is: "*Compact dual-wavelength depolarization lidar for aerosol characterization over the Subtropical North Atlantic*".

*Comment # 2:* Line 189 and following: You retrieved (columnar) lidar ratios iteratively by fitting your (whole/or parts of) lidar profile to AERONET AOD, which is a sound method. However, I recommend to be cautious or even to refrain from calling the lidar ratios "measured" or "observed" (e.g., Line 575, Line 606). In Line 321 (and some more), you call it "effective LR retrieved", which I believe is more appropriate. I recommend to do it in this way consistently throughout the manuscript.

*Reply*: To be consistent throughout the manuscript we have decided to call the *LR* as *retrieved LR*. The changes were made in the following lines

L189: we changed "The LR is iterated" for "We effectively retrieved the LR by iteration"

L232, 269, 388, 472, 475, 477, 528, 532, 539, 544, 546, 568, 577: we changed "LR" for "retrieved LR"

L322, 331, 379, 391, 398, 520: we changed "effective LR retrieved" for "retrieved LR"

L441: we changed "lidar ratio (LR)" for "retrieved LR"

**Comment # 3:** In Line 191, you say that daylight impairs 808 nm detection above 2 km a.g.l., and that you solve this by forward Klett. How about 532 nm? Is it totally free of this impairment? Or does it just occur at higher altitudes as well (which would less impact your retrieval)? Also, in Line 193, concerning the estimated AOD at 808 nm, I think this needs to be explained more clearly, or a reference to Sanchez-Barrero et al. (2024) has to be given (like in Line 380–381).

*Reply*: Both the 532 nm and 808 nm channels are affected by solar background during daytime. However, the narrower field of view and higher laser pulse energy of the 532 nm channel enable it to achieve a higher detection limit compared to the 808 nm channel, resulting in less severe impairment. At 532 nm, this impact generally occurs at higher altitudes, where the retrieval process is less affected. Regarding the estimated AOD at 808 nm, it is thoroughly explained in Sanchez-Barrero et al. (2024). This is now better explained at the end of the first paragraph in Sect. 2.1.2 *Derived aerosol properties*

**Comment # 4:** Line 224–233: This whole section rather belongs to the uncertainty discussion (Line 171 following).

*Reply*: We agree and have made the necessary changes. The structure of the paper has been rearranged to better align with the scope. The two paragraphs previously located in lines 174–182 and 226–235 have now been combined to form the final paragraph of Section 2.1.2, titled *Derived Aerosol Properties*.

**Comment # 5:** Line 231–232: The sentence "The convergence of the AOD leads then to the uncertainties of the LR." is not clear enough to me. How does this work? Or is this explained in detail in Sanchez-Barrero et al. (2024)? Then add that reference and a sentence stating that there as well.

*Reply*: The uncertainty into the LR is roughly estimated by the convergence within the AOD uncertainties (0.01) in the iterative Klett solution (Sect. 3.2 in Sanchez-Barrero et al., 2024). This explanation and the reference to Sanchez-Barrero et al. (2024) is now included at the end of the second paragraph in Sect. 3.1.

**Comment # 6:**

Line 269 (and others): Do the reported ±-ranges include uncertainties of the individual instruments? What is the reference? I guess, the MPL? So, the values of the CIMEL are on average all smaller?

*Reply*: As explained in the manuscript, when we describe the uncertainties of a given parameter for each instrument individually, the reported ±-ranges account for the uncertainties of the individual instruments. Only in the second-to-last paragraph of Sect. 3.1, when discussing absolute differences while comparing CE376 and MPL-measured parameters, does the absolute difference refer to the mean absolute difference over a given time frame. Specifically, it represents the difference between the 15-minute average signals, considering both the first 4 km of the profile (up to 6.5 km a.s.l.) and the complete dataset. Here, as guessed by the referee, the reference is MPL and the values of the CIMEL tend to be on average smaller.

**Comment # 7:**

In general, when speaking about Cumbre Vieja sulphate, it would be valuable to cite Gebauer et al. (2024, https://doi.org/10.5194/acp-24-5047-2024), especially with regards to lidar ratios (of $66.9 \pm 10.1$, $60.2 \pm 9.2$ and $30.8 \pm 8.7$ sr at 355, 532 and 1064 nm) reported by them, which were directly measured by Raman lidar.

*Reply*: We thank the referee for bringing this work to our attention. The article has now been incorporated into the discussion. The last sentence of the fourth paragraph (addressing the first time frame studied) reads as follows:

"At Cabo Verde, south of the volcano, Raman lidar measurements at 532 nm on September 24, 2021, revealed a similarly enhanced lidar ratio of $60.2 \pm 9.2$ sr within the sulfate-dominated planetary boundary layer where the sulfate mass concentration was $133.1 \pm 20.3$ µg m$^{-3}$). The measurements indicated a higher $\alpha a$ (532) of $549 \pm 38$ Mm$^{-1}$ and $\delta p$ (532) of $0.7 \pm 1.0$, suggesting a predominance of fine-mode aerosols within a more aerosol-loaded layer (AOD of $0.43 \pm 0.02$). These values, which exceeded those measured at Izaña, suggest a stronger influence of the sulfuric volcanic plume at Cabo Verde during that day (Gebauer et al., 2024) (Fig. 3h)."

The citation is also included in the references.

**Minor comments:**

**Comment # 8:** Line 25–26: I would avoid to report volume depolarization ratios in %, but rather in floating point numbers. This then also avoids confusion about absolute and relative differences.

*Reply*: the correction has been made to now report it in floating numbers.

**Comment # 9:**

Line 55: "cloud altitude", I suggest to write cloud base height.

*Reply*: Corrected

**Comment # 10:**

After Line 95, a short paragraph summarizing the content of the manuscript could be added to guide the reader about a little bit. Something like "In Sect. xyz, we describe this and that. The results are shown in Sect. xyz".

*Reply*: Following the suggestion of the second referee, we have reorganized the manuscript structure to better align with its scope. The final paragraph of the Introduction now outlines the content of the manuscript as follows

"Section 2 outlines the instrumentation and methodology used to derived columnar aerosol properties. Section 3 presents the results, with Sect. 3.1 detailing the intercomparison campaign, and Sect. 3.2 exploring aerosol properties under different scenarios: Saharan dust (Sect. 3.2.1), volcanic aerosols (Sect. 3.2.2) and wildfire aerosols (Sect. 3.2.3). Sect. 3.3 summarizes the observations, proving an aerosol classification. The main conclusions of this study are summarized in Sect. 4."

**Comment # 11:** Line 100–102: Is the laser at 808 nm also eye-safe? And Line 655–656: would it be still eye-safe after that upgrade?

*Reply*: Yes, the 808 nm channel is eye-safe. This has now been explicitly stated in the system description (Sect. 2.1). The intention is to maintain the eye-safe mode for this compact model following the upgrade. We have modified the text slightly to try to make it clearer.

***Comment # 12:***

Line 176: "AOD<0.03". From lidar or photometer? At which wavelength?

*Reply*: We apologize for not including this info. The AOD presented here is measured by the photometer at 500 nm. The wavelenght is now included in the manuscript.

***Comment # 13:***

Line 204: The software iAAMS, is it open source? Is it part of the CE376 CIMEL micro-lidar package? Can it be cited?

*Reply*: The iAAMS software is not open source. It is available upon request as part of the CE376 CIMEL micro-lidar package and supports the processing of data from both CIMEL photometers and lidar systems. However, no specific publication about the software is currently available.

***Comment # 14:***

Also Lines 655–656: "Upcoming developments on the CE376 CIMEL micro-lidar aim to upgrade both lasers, the visible laser (532 nm) to a more powerful one, and the near-infrared laser (808nm) to reduce its sensitivity to solar background during daylight." This sentence is not well comprehensible (due to imho wrong comas and position of parts of the sentence). The sensitivity to solar background during daylight is not a property of the laser but the detectors/field-of-view. Of course, stronger lasers help in detecting signals also in daylight contaminated periods of the day. Please, change the sentence.

*Reply*: We do agree with the referee. The first two sentences of this paragraph has been modified for clarification as follows: "Upcoming developments for the CE376 aim to upgrade both lasers to more powerful, eye-safe models, enhancing the system's vertical resolution and capabilities, optimizing its operation. The use of two wavelengths will provide more precise information on the size, composition, vertical distribution, and temporal evolution of atmospheric aerosols."

***Comment # 15:***

In most figures: Is there really a.g.l. and not a.s.l. meant? I believe the latter. I recommend to call the y-axis always the same (Z, but better altitude or even height) with always the same meaning (a.s.l. or a.g.l.) and unit (m or km (not Km)).

*Reply*: We thank the referee for noticing the mistakes. In Figures 1-8 we have modified the y-axis label to be homogenously called "altitude" and corrected the reference level to "above sea level (m a.s.l.)". Time frame has been also corrected by adding UTC in Figures 2, 4, 5, 6 and 8.

**Minor technical and spelling comments:**

***Comment # 16:*** Generally, write "Fig." when you cite a figure in a sentence. Only use "Figure" at the very beginning of a sentence. The same is true for Section (Sect.) and Equation (Eq.).

*Reply*: we have changed the citation of figures (to Fig.) and sections (to Sect.) and equations (to Eq.) accordingly when citing.

***Comment # 17:*** Generally, try to use a single, consistent date format (everywhere, in text, figures and captions). I suggest as an example: "15 December 1965" or "December 15, 1965". In Line 493 for

example, it is suddenly different. Furthermore, when you give a timestamp, always add UTC (I believe it is UTC everywhere).

*Reply*: Dates have been homogenised to the month day, year format. We have also included the UTC reference when time stamp is used.

**Comment # 18:**

Generally, use – (--) hyphen for intervals. Either consistently use spaces before and after plus-minus signs or not, for example 5~$\pm$~6 (as mostly in the manuscript) or only 5$\pm$6 (as in Sect. 5).

*Reply*: We have corrected the manuscript and now hyphens and dashes are used properly. We have removed the spaces before and after the plus-minus signs.

**Comment # 19:**

Figure 1: The sentence in the caption "Reference: Google (2023) Montaña de Izaña. Imágenes ©2023 CNES / Airbus, 252 GRAFCAN, Maxar Technologies [accessed on January 31, 2024]" is not a particular help if one wants to find this, even though it is much information/text. Better provide a clear link and add "adapted".

*Reply*: Figure 1f corresponds to a photo taken by Pekka Pelkonen from ICOS RI. This is now specified in the caption at follows: "f) Location of the CE376 and MPL44161 lidar systems at Montaña de Izaña. uring the comparison campaign carried out in August 2021. Photo courtesy of Pekka Pelkonen (The Integrated Carbon Observation System, ICOS RI)."

**Comment # 20:** Also in Fig. 1, the sentence "Yellow dashed bars in (a)-(e) highlight the time frames selected to calculate the averaged profiles described in Figure 2." has to be placed before f) and g) are described.

*Reply*: We do agree, and correction has been made accordingly.

**Comment # 21:**

Line 10: In "F-59000", delete "F-", as you also do not add it in Line 8 ("Paris").

*Reply*: Corrected

**Comment # 22:**

Line 12: Add USA in the end, as you added France and Spain as well above.

*Reply*: Added

**Comment # 23:** Line 13: What is TRAGSATEC? Is it an abbreviation? Write it out.

*Reply*: TRAGSATEC is a company name. The company has already been part of previous publications in AMT and ACP. Some examples are shown below. The zip code has been added.

Atmos. Chem. Phys., 22, 739–763, 2022
https://doi.org/10.5194/acp-22-739-2022
[CC BY]

Atmospheric
Chemistry
and Physics

[Figure]

**Long-term characterisation of the vertical structure of the Saharan Air Layer over the Canary Islands using lidar and radiosonde profiles: implications for radiative and cloud processes over the subtropical Atlantic Ocean**

África Barreto[1,2], Emilio Cuevas[1], Rosa D. García[3,1], Judit Carrillo[4], Joseph M. Prospero[5], Luka Ilić[6], Sara Basart[7], Alberto J. Berjón[3,1], Carlos L. Marrero[1], Yballa Hernández[1,8], Juan José Bustos[1], Slobodan Ničković[9], and Margarita Yela[10]

[1]Izaña Atmospheric Research Center (IARC), Agencia Estatal de Meteorología (AEMET), Santa Cruz de Tenerife, Spain
[2]Atmospheric Optics Group of Valladolid University (GOA–UVa), Valladolid University, Valladolid, Spain
[3]TRAGSATEC, Madrid, Spain

Atmos. Chem. Phys., 22, 11105–11124, 2022
https://doi.org/10.5194/acp-22-11105-2022
[CC BY]

Atmospheric
Chemistry
and Physics

[Figure]

Research article

**Aerosol characterisation in the subtropical eastern North Atlantic region using long-term AERONET measurements**

África Barreto[1,2], Rosa D. García[3,1], Carmen Guirado-Fuentes[2,1,a], Emilio Cuevas[1], A. Fernando Almansa[4,1], Celia Milford[1], Carlos Toledano[2], Francisco J. Expósito[5], Juan P. Díaz[5], and Sergio F. León-Luis[3,1]

[1]Izaña Atmospheric Research Center (IARC), Agencia Estatal de Meteorología (AEMET), Santa Cruz de Tenerife, Spain
[2]Atmospheric Optics Group of Valladolid University (GOA–UVa), Valladolid University, Valladolid, Spain
[3]TRAGSATEC, Madrid, Spain

[Figure]

Article

**Spectral Aerosol Radiative Forcing and Efficiency of the La Palma Volcanic Plume over the Izaña Observatory**

[Figure]

Rosa Delia García[1,2], Omaira Elena García[2], Emilio Cuevas-Agulló[2,*], África Barreto[2], Victoria Eugenia Cachorro[3], Carlos Marrero[2], Fernando Almansa[4], Ramón Ramos[2] and Mario Pó[5]

[1]  TRAGSATEC, 28037 Madrid, Spain

**Comment # 24:**

Line 67: "Laboratoire d'optique atmosphérique" with capital letters like in Line 10. This whole sentence concerning software and the joint laboratory could be rephrased or moved to some other place, where it fits better (maybe along iAAMS in Line 204?).

*Reply*: We do agree. Capital letters and the sentence have been moved to the last paragraph of section 2.3 where the iAAMS software is introduced.

*Comment # 25:*

Line 72–73 before unit hPa add non-line-breaking space (~ or \,).

*Reply*: The sentence has been shortened to avoid the line-breaking

*Comment # 26:*

Line 86: You use MPLNET before you introduce it (in Line 207–208).

*Reply*: We now introduce MPL in the second last paragraph of the introduction and rephrased the first sentence of Sect. 2.3 to avoid repetition.

*Comment # 27:*

Line 146–147: EARLINET abbreviation is not given. Last access date of url is not given. Furthermore, I suggest to cite Pappalardo et al. (2014, https://doi.org/10.5194/amt-7-2389-2014) in this context.

*Reply*: We thank the referee for citation suggested. We defined the acronym, added the last accessed date and included the citation.

*Comment # 28:*

Line 147–148: "RCS ($\lambda$,r)" definetly should not line-break.

*Reply*: The line break is corrected.

*Comment # 29:*

Line 160: I suggest to call it "$\Delta 90°$ method" instead of "$\pm 45°$ method", and also cite Freudenthaler et al. (2016). Or like Papetta et al. (2024, https://doi.org/10.5194/amt-17-1721-2024) who managed it this way: "The depolarization calibration suggested by the manufacturer follows the $\Delta(\pm 45°)$ method described in Freudenthaler et al. (2009), later renamed to $\Delta(90°)$ (Freudenthaler, 2016)."

*Reply*: We agree with the referee's suggestion, and the method has been renamed as stated in Freudenthaler (2016).

*Comment # 30:*

Line 178 and 228: "kilometer" instead of "kilometre".

*Reply*: Correction done.

*Comment # 31:*

Line 238–239: add non-line-breaking space between wavelength (532, 808) and unit (nm).

*Reply*: Correction done.

*Comment # 32:*

Line 241: MODIS instead of Modis, add last access date to url, or better not to give url in text at all, but only in the caption of Fig. 1.

*Reply*: MODIS is now written in capital letters, more specifically as MODIS VIS channel and the url and access to it is included in Fig 1 and deleted in the text.

*Comment # 33:*

Line 253: and last access date to url.

*Reply*: Included

*Comment # 34:*

Line 269: "21h- 6h" what is this? Local time? UTC? I would write "21:00-06:00 UTC" or even better night-time.

*Reply*: Corrected to UTC format

*Comment # 35:* Line 270-271: non-line-breaking space before km.

*Reply*: The sentence has been slightly modified to avoid the line breaking.

*Comment # 36:*

Line 293: Call it everywhere the same "extinction Ångström exponent".

*Reply*: The correction has been made

*Comment # 37:*

Line 299: "In this section," comma.

*Reply*: Corrected

*Comment # 38:*

Line 311–312: The abbreviation SAL does not need to be explained again (it is already explained in Line 306). So just use SAL there.

*Reply*: Corrected

*Comment # 39:*

Line 318, 326, 333, 378, 390, 397, 518, 524, 529, 535: I do not understand the underscoring. Avoid it.

*Reply*: The underscore has been removed.

*Comment # 40:* Line 320: 808 non-line-breaking space nm.

*Reply*: Corrected

*Comment # 41:*

Line 329–330: non-line-breaking space before unit.

*Reply*: Corrected

*Comment # 42:*

Line 333: Do not call it "the arrival of a new episode" but rather "arrival of a new dust layer", "a new dust episode occurred", or something like that.

*Reply*: The sentence has been corrected.

**Comment # 43:**

Line 336–337: The unit has to be non-line-breaking as well (-1).

*Reply*: Corrected.

**Comment # 44:**

Line 358–359: non-line-breaking space before unit.

*Reply*: Corrected.

**Comment # 45:**

Line 370: last access date to url.

*Reply*: Last access has been included.

**Comment # 46:**

Line 367: "Image" or "Photo" instead of "Imagen".

*Reply*: Corrected

**Comment # 47:**

Line 385–386: no space between number and %, or \, which Copernicus uses, I believe.

*Reply*: Corrected

**Comment # 48:**

Figure 5h and 6h. UTC? Date format?

*Reply*: Time frame has been also corrected by adding UTC in Figures 2, 4, 5, 6 and 8.

**Comment # 49:**

Line 400–401: non-line-breaking spaces before unit.

*Reply*: Corrected

**Comment # 50:**

Line 406: "slight" instead of "slightly".

*Reply*: Corrected

**Comment # 51:**

Line 440–441: non-line-breaking spaces before unit.

*Reply*: Corrected

***Comment # 52:***

Line 441–442: non-line-breaking spaces before figure number.

*Reply*: Corrected

***Comment # 53:***

Line 444: (non-line-breaking) spaces before units.

*Reply*: Corrected

***Comment # 54:***

Line 476: non-line-breaking spaces before unit

*Reply*: Corrected

***Comment # 55:***

Line 490, 491, 550, 551: Use μm instead of micron.

*Reply*: Corrected

***Comment # 56:***

Line 551,552: Omit \cdot in units, as it is also not used in Mm-1 sr-1 for example.

*Reply*: Corrected

***Comment # 57:***

Line 575–576: non-line-breaking spaces before unit.

*Reply*: Corrected

***Comment # 58:***

Line 586: μm instead of um.

*Reply*: Corrected

***Comment # 59:***

Line 629: "(0-24h)", again, I would avoid that, just write the "full-day dataset" or something similar.

*Reply*: We removed the time arrange and rephrased using "full-day dataset".

***Comment # 60:***

Line 656: non-line-breaking spaces before unit.

*Reply*: Corrected

***Comment # 61:***

Line 657: For GRASP, a citation has to be added.

*Reply*: The citations to Dubovik et al. (2021 and 2014) have been added.

**References:**

***Comment # 62:*** Generally, use either standard abbreviations (as is in the end standard at Copernicus) or full journal names.

*Reply*: the references have been reviewed to use standard abbreviations.

***Comment # 63:***

Generally, use a consistent doi format. Standard is with https://.

*Reply*: the doi format has been revised and corrected in the references

***Comment # 64:***

Generally, between initials of the first names, there needs to be a space (at Copernicus).

*Reply*: Corrections have been made accordingly

***Comment # 65:***

Line 710: last access dates have to be given.

*Reply*: last access has been included

***Comment # 66:***

Line 758: dois do not need a last access date.

*Reply*: the last access has been removed

***Comment # 67:*** Line 759: Cuevas et al. (2021): Would be nice to have an url (with last access date): https://www.aemet.es/documentos/es/conocermas/recursos_en_linea/publicaciones_y_estudios/public aciones/GAW_Report_No_259/GAW_Report_No_259.pdf

*Reply*: The url and last access date have been included.

***Comment # 68:*** Line 803: García et al. (2014): Pages are wrong (besides strange doi format). It should be 179–194. Furthermore, there should be a colon (:) after "station" in the title.

*Reply*: The reference has been corrected.

***Comment # 69:***

Line 806–809: Gasteiger and Freudenthaler (2014) is somehow mixed up with some chapter of IPCC.

*Reply*: The reference has been corrected.

**Comment # 70:**

Line 843: Kusmierczyk-Michulec et al. (2002) has three authors, not only one.

*Reply*: Corrected

**Comment # 71:**

Line 867–871: Milford et al. (2023): There are strange semicolons in the author list. Furthermore, ISSN is not needed in a journal and the year (2023) usually does not need to be given twice.

*Reply*: The semicolons have been replaced by comas and the ISSN removed.

**Comment # 72:**

Line 899–900: In Rodríguez et al. (2015), not "Löpez-Solano" but "López-Solano" and in the title not "North Africa dipol" but "North African dipol".

*Reply*: the correction has been made

**Comment # 73:**

Line 928–937: All three Tesche et al. (2009a,b,2011) citations have strange format (a missing year, "63:4", "vol.").

*Reply*: The format of the three Tesche articles have been corrected.

**Reviewer 2**

*"The paper presents a comparative performance assessment of two collocated lidar systems in various atmospheric aerosol loading scenarios over the Izaña site in the North Atlantic. Although both systems were described in previous papers, the added value brought by this work is considered relevant and suitable for publication in AMT after the authors address the following minor revisions:"*

**General comments**

**Comment # 1:** The structure of the paper could be improved for a better understanding by the readers of the scope of the paper. A clearer structure, in line with the twofold motivation presented, with minimal effort from the authors' part, could be something like:

Introduction
Instrumentation and methodology
Results and discussions
Intercomparison between the CE376 and MPL-4B
Distribution of aerosol properties derived from CE376
Saharan dust
Volcanic
Aerosol classification
Conclusions
Bibliography

*Reply*: We agree with the referee that rearranging the structure of the paper would make it easier to follow its scope. The revised structure is as follows:
1. Introduction
2. Instrumentation and methodolody
      2.1 The CE376 lidar
          2.1.1 Signal processing and calibrations
          2.1.2 Derived aerosol properties
      2.2 The CE318-T photometer
      2.3 The MPL-4B lidar
3. Results and discussion
      3.1 Intercomparison between the CE376 and MPL-4B
      3.2 Distribution of aerosol properties derived from CE376
          3.2.1. Saharan dust aerosols
          3.2.2 Volcanic aerosols
          3.2.3 Wildfire aerosols
              3.2.3.1 Aged aerosols from the Canadian wildfires
              3.2.3.2 Fresh aerosols from forest fires at Tenerife
      3.3 Aerosol classification
4. Conclusions

**Comment # 2:** Please have wording and phrasing checked by a native English speaker.

*Reply*: We have revised the language in the manuscript to conform to American English standards.

**Comment # 3:** Please check all references for the correct citation form.

*Reply*: the references have been reviewed to use standard abbreviations. We have also corrected the doi formats and we have included spaces between initials of the first names.

**Particular comments**

***Comment # 4:*** Lines 1-2: Rephrase the title: introduce the idea that there are 2 lidar systems in discussion; the use of "North Atlantic aerosols" might induce to readers that it's a new class of aerosols, which is not the case; "…a compact dual-wavelength depolarization Lidar Observations", either you delete "a" or you delete the plural from" Observations"

*Reply*: We agree with the reviewer and have removed the capitalization and modified the title to make it more appropriate. The new tittle is: "*Compact dual-wavelength depolarization lidar for aerosol characterization over the Subtropical North Atlantic*".

***Comment # 5:*** Line 13: Please detail the abbreviation TRAGSATEC, provide address, etc.

*Reply*: TRAGSATEC is a company name. The company has already been part of previous publications in AMT and ACP. Some examples are shown below. The zip code has been added.

Atmos. Chem. Phys., 22, 739–763, 2022
https://doi.org/10.5194/acp-22-739-2022

[Figure]

Atmospheric
Chemistry
and Physics

**Long-term characterisation of the vertical structure of the Saharan Air Layer over the Canary Islands using lidar and radiosonde profiles: implications for radiative and cloud processes over the subtropical Atlantic Ocean**

África Barreto[1,2], Emilio Cuevas[1], Rosa D. García[3,1], Judit Carrillo[4], Joseph M. Prospero[5], Luka Ilić[6], Sara Basart[7], Alberto J. Berjón[3,1], Carlos L. Marrero[1], Yballa Hernández[1,8], Juan José Bustos[1], Slobodan Ničković[9], and Margarita Yela[10]

[1]Izaña Atmospheric Research Center (IARC), Agencia Estatal de Meteorología (AEMET), Santa Cruz de Tenerife, Spain
[2]Atmospheric Optics Group of Valladolid University (GOA–UVa), Valladolid University, Valladolid, Spain
[3]TRAGSATEC, Madrid, Spain

Atmos. Chem. Phys., 22, 11105–11124, 2022
https://doi.org/10.5194/acp-22-11105-2022
Atmospheric
Chemistry
and Physics

Research article

**Aerosol characterisation in the subtropical eastern North Atlantic region using long-term AERONET measurements**

África Barreto[1,2], Rosa D. García[3,1], Carmen Guirado-Fuentes[2,1,a], Emilio Cuevas[1], A. Fernando Almansa[4,1], Celia Milford[1], Carlos Toledano[2], Francisco J. Expósito[5], Juan P. Díaz[5], and Sergio F. León-Luis[3,1]

[1]Izaña Atmospheric Research Center (IARC), Agencia Estatal de Meteorología (AEMET), Santa Cruz de Tenerife, Spain
[2]Atmospheric Optics Group of Valladolid University (GOA–UVa), Valladolid University, Valladolid, Spain
[3]TRAGSATEC, Madrid, Spain

[Figure]

*remote sensing*

MDPI

*Article*
**Spectral Aerosol Radiative Forcing and Efficiency of the La Palma Volcanic Plume over the Izaña Observatory**

Rosa Delia García[1,2], Omaira Elena García[2], Emilio Cuevas-Agulló[2,*], África Barreto[2], Victoria Eugenia Cachorro[3], Carlos Marrero[2], Fernando Almansa[4], Ramón Ramos[2] and Mario Pó[5]

[1] TRAGSATEC, 28037 Madrid, Spain

**Comment # 6:** Line 22: Consider using "while the systems were collocated"

*Reply*: The suggestion has been included.

**Comment # 7:** Lines 31-32: When discussing fine mode, consider separating the values for Cumbre Vieja and Canadian wildfires

*Reply*: The two aerosol types are now indicated separately as follows: "In contrast, smaller particles with quasi homogeneous morphology were attributed to sulfate aerosols from the early stages of the Cumbre Vieja volcano eruption and aged Canadian wildfire plumes traveling across the Atlantic. These aerosols showed the lowest $\delta^P$ (0.03 for volcanic sulfate and 0.08 for aged wildfire aerosols) and the highest EAE (532/808) (1.5 and 1.2, respectively)."

**Comment # 8:** Line 34: Given the limitations discussed in the paper, consider replacing "excellent performance" with "suitability"

*Reply*: We have replaced the wording as suggested.

**Comment # 9:** Line 42: Consider using "Ground-based remote sensing instruments…"

*Reply*: Correction done.

**Comment # 10:** Line 80: Detail "These events". What events are you referring to?

*Reply*: We agree with the referee that the sentence is not presented clear. We were refereeing to the background clean conditions. We have rephrased a few sentences to clarify the specific events we are referencing. The paragraph now reads as follows: "This study was conducted at the Izaña Atmospheric Research Centre (IARC, 28.31° N, 16.50° W), a multiplatform site providing long-term measurements of atmospheric chemical and aerosol species (Cuevas et al., 2024), operated by the State Meteorological Agency of Spain (AEMET). The observatory is also part of ACTRIS (Aerosol, Clouds and Trace Gases Research Infrastructure) European Research Infrastructure (Laj et al., 2024) as a Central Facility for aerosol remote sensing. Located at 2,367 meters above sea level (average pressure of 770 hPa), in the vicinity of Teide National Park (Canary Islands), the site benefits from stable atmospheric conditions largely governed by the quasi-permanent northwest subsidence regime of the Hadley cell, and well-stratified lower troposphere,  with strong temperature inversion layer often situated between 900 and 800 hPa (Cuevas, 1995; Carrillo et al., 2016). Background conditions prevail for most of the year, occurring more frequently between April and June. In contrast, dust-loaded Saharan air masses dominate in July and August (Barreto et al.,2022b). These alternating conditions make Izaña an ideal location for studying regional aerosol transport in the subtropical North Atlantic."

**Comment # 11:** Line 86: You already used the CE376 abbreviation above. Introduce it when first used.

*Reply*: Changes have been made to introduce it when first used. Same for MPL-4B.

**Comment # 12:** Lines 88-91: Mention also the use of the CE-318T photometer

*Reply*: We have decided not to mention the CE-318T photometer in this particular line, as it is used in this study primarily to improve the retrieval of the AOD profile from the lidar systems. In this paragraph, we focus on the lidar retrievals of additional aerosol metrics, such as $\delta^P$ (532, 808), ACR (808-532) and EAE (532/808).

***Comment # 13:*** Lines 92-95: This is a bit redundant placed here, you stated this at lines 39-47. Consider removing this paragraph.

*Reply*: We do agree with the referee and the parrapgrah has been removed. Instead we included a brief descrption of the content of each section.

***Comment # 14:*** Lines 97-204: Consider including in this section a paragraph discussing advantages and disadvantages of the lidar systems used in this paper compared to high-power lidars.

*Reply*: The CE376 lidar system used in this study is a compact, lightweight, and eye-safe instrument designed for continuous monitoring with minimal maintenance requirements. Unlike high-power lidars, which often require specialized laboratories, frequent maintenance interventions, and are more susceptible to adverse meteorological conditions, the CE376 is versatile and easier to deploy in field settings. However, a notable limitation of the CE376 is its reliance on photometer data to retrieve a vertically constant lidar ratio (LR). This contrasts with high-power Raman or HSRL (High Spectral Resolution Lidar) lidar systems, which can retrieve vertically resolved LR profiles independent of photometer information, providing higher vertical resolution and accuracy in some applications specially during night time (Weitkamp, 2005; Ansman et al., 1990; Shipley et al., 1983).

This explanation has been included along section 2.1, together with the new to references (Ansman et al., 1990 and Shipley et al., 1983)

***Comment # 15:*** Line 97: Consider adding the CE-318T in the section title, given the fact that you discuss it extensively at lines 117-126.

*Reply*: We introduced the CE318-T photometer in line 65 of Sect. 1. A detailed description is provided later in Sect. 2, "Instrumentation and Methodology" (Sect. 2.2).

***Comment # 16:*** Lines 102-103: Is the 808nm also eye-safe?

*Reply*: Yes, the 808 nm channel is eye-safe (Sect. 2.1).

***Comment # 17:*** Lines 109-110: Add a citation for the technical data of the electronic cards, if available.

*Reply*: The electronic system includes single-photon counting modules operating at rates of up to 40 MHz, an FPGA for multichannel sequencing and counting implemented with internally developed HDL code, and a microcontroller-based board for laser power, temperature, and security control and monitoring. Additionally, it features a fully digital power and timing control system, as well as a USB-based communication interface. These details are now included at the end of the first paragraph in Sect. 2.1. Unfortunately, there are no publications currently available related to this work.

***Comment # 18:*** Line 146: Describe the acronym EARLINET (first used here).

*Reply*: We defined the acronym and included a citation to Pappalardo et al. (2014).

***Comment # 19:*** Line 161-163: Why did you not perform more frequent +/- 45 polarization calibrations?

*Reply*: The version of the CE376 lidar system used in this work required manual polarization calibrations using a half-wave plate with a precision of two degrees. This manual approach increased the probability of human error and risked displacing optical components, potentially affecting the system's overall stability. To ensure the system configuration remained stable, polarization calibrations were only performed during necessary maintenance interventions. Importantly, the latest version of the

CE376 lidar system now includes a motorized calibration mechanism, significantly improving calibration precision and ease of operation, allowing for more frequent and reliable calibrations.

**Comment # 20:** Line 176: AOD<0.03 @ what wavelength?

*Reply*: We apologize for not including this info. The AOD is measured by the photometer at 500 nm. The wavelenght is now included in the manuscript.

**Comment # 21:** Lines 191-193: Please elaborate on the solar background limitation during daytime, and why it's different between the 532nm and 808nm channels.

*Reply*: During daytime operations, the 808 nm channel is more susceptible to solar background noise due to their larger field of view and the lower pulse energy of the emitted laser beam compared to the 532 nm channels. The increased field of view results in a greater collection of ambient solar radiation, which combined with the lower pulse energy, reduces the signal-to-noise ratio. In contrast, the 532 nm channels benefit from a narrower field of view and higher pulse energy, making them less affected by solar background under similar conditions.

|  | CIMEL CE376 GPNP |  |
|---|---|---|
| Wavelength | 532 nm | 808 nm |
| Laser source | Frequency doubled Nd:YAG | Pulsed laser diode |
| Pulse energy | 5-10 uJ (15-20 μJ) | 3-5 uJ |
| Repetition rate (Pulse width) | 4.7 kHz (20 ns) | 4.7 kHz (186 ns) |
| Emission/Reception (E/R) | Biaxial | Biaxial |
| Telescope (E/R) | Galilean | Galilean |
| Diameter (E/R) | 100 mm / 100 mm | 100 mm / 100 mm |
| Half Field of View (E/R) | 100 μrad / 120 μrad | 240 μrad / 330 μrad |
| Depolarization | Yes | Yes |

The last 3 sentences of the first parragraph in the new Sect. 2.1.2 has been modified as follows: "During daytime operations, the 808 nm channel is more susceptible to solar background noise due to their larger FOV and the lower pulse energy of the emitted laser beam compared to the 532 nm channel. The increased FOV leads to a greater collection of ambient solar radiation, and when combined with the lower pulse energy, reduces the signal-to-noise ratio. This limitation lowers the detection capability of the 808 nm channel to below 2 km above ground level. To address this, the forward solution is employed, and the inversion constrained using an estimated AOD for the reduced profile at 808 nm (Sanchez-Barrero et al., 2024). In contrast, the 532 nm channel benefits from a narrower FOV and higher pulse energy, making it less affected by solar background under similar conditions. This allows the 532 nm channel to achieve a higher detection limit compared to the 808 nm channel. Consequently, the impact of solar background noise on the 532 nm channel is less severe and typically occurs at higher altitudes, where the retrieval process is less affected.

**Comment # 22:** Line 206: Is this a proper validation process? "Intercomparison of the aerosol products of the CE376 and MPL-4B" would be more suitable.

*Reply*: We do agree with the referee. The section, now Sect. 3.1, has been renamed to "Intercomparison between the CE376 and MPL-4B"

*Comment # 23:* Line 241: Rephrase "the grade of impact" with "spatial extent of the Saharan dust layer(s) over…"

*Reply*: the correction has been made.

*Comment # 24:* Line 241: Replace "Modis" with "MODIS VIS channel"

*Reply*: the correction has been made.

*Comment # 25:* Line 242: "The additional information given by the 808 nm channel of the CE376 is later discussed in more detail in section 4.1." Either expand a bit here or remove this sentence.

*Reply*: The sentence has been removed to avoid repetition.

*Comment # 26:* Line 244: Increase the size of the photometer data and legend at e). It can be hard to read. This is valid for all subsequent similar plots.

*Reply*: The size in the AOD plots have been increased.

*Comment # 27:* Lines 252-253: Rephrase "the grade of impact" with "spatial extent of the Saharan dust layer(s) over…"; Replace "Modis" with "MODIS VIS channel".

*Reply*: the suggested corrections have been made.

*Comment # 28:* Line 289: Consider replacing "catching the impact of" with "depicting";

*Reply*: corrected

*Comment # 29:* Line 312: for dust - oil smoke mixes have a look at: https://doi.org/10.5194/acp-22-5071-2022

*Reply*: We thank the referee for bringing this work to our attention. The reference has been included into the discussion.

*Comment # 30:* Lines 406-408: Please rephrase. The wording can be confusig. Avoid using "likely", "slightly". If you decide to keep it, it shoule be: "This is in slight contrast"

*Reply*: The word "likely" has been removed as it is redundant.

*Comment # 31:* Line 417, Line 428, Line 431: As mentioned above, consider replacing "Impact" when describing aerosol optical properties.

*Reply*:

Line 417: we have replaced "Impact of the Cumbre Vieja volcanic plume in the aerosol properties measured at Izaña expressed as average aerosol profiles measured during September 23–24, 2021" by "Aerosol properties of the Cumbre Vieja volcanic plume measured at Izaña expressed as average aerosol profiles measured during September 23-24, 2021".

Line 428: we have replaced "impacted" by "reached"

Line 431: we have replaced the sentence "These days were selected to avoid or minimize the simultaneous impact of Saharan dust particles." by "These days were chosen to minimize or avoid simultaneous interaction with Saharan dust particles."….etc

**Comment # 32:** Lines 456-457: For consistency, mention MODIS VIS channel as above.

*Reply*: corrected

**Comment # 33:** Line 575: Add "retrieved" before LR.

*Reply*: corrected

**Comment # 34:** Lines 596-597: Please rephrase: "In contrast, better constrained lidar ratios using sun photometers, alone, still not good for aerosol typing when using alone".

*Reply*: we have rephrased the sentence as follows: "In contrast, while sun photometers can help constrain lidar ratios more effectively, these ratios are still insufficient for aerosol typing when using alone."